# Skin immunisation activates an innate lymphoid cell-monocyte axis regulating CD8$^+$ effector recruitment to mucosal tissues

Marija Zaric [1], Pablo D. Becker [1], Catherine Hervouet[1], Petya Kalcheva[1], Andor Doszpoly[2], Negin Blattman[3], Lauren A. O' Neill[1], Barbara Ibarzo Yus[1], Clement Cocita[1], Sung-Yun Kwon[4], Andrew H. Baker[2], Graham M. Lord [1,5] & Linda S. Klavinskis [1]

CD8$^+$ T cells provide a critical defence from pathogens at mucosal epithelia including the female reproductive tract (FRT). Mucosal immunisation is considered essential to initiate this response, however this is difficult to reconcile with evidence that antigen delivered to skin can recruit protective CD8$^+$ T cells to mucosal tissues. Here we dissect the underlying mechanism. We show that adenovirus serotype 5 (Ad5) bio-distributes at very low level to non-lymphoid tissues after skin immunisation. This drives the expansion and activation of CD3$^-$ NK1.1$^+$ group 1 innate lymphoid cells (ILC1) within the FRT, essential for recruitment of CD8$^+$ T-cell effectors. Interferon gamma produced by activated ILC1 is critical to licence CD11b$^+$Ly6C$^+$ monocyte production of CXCL9, a chemokine required to recruit skin primed CXCR3$^+$ CD8$^+$T-cells to the FRT. Our findings reveal a novel role for ILC1 to recruit effector CD8$^+$ T-cells to prevent virus spread and establish immune surveillance at barrier tissues.

[1] School of Immunobiology and Microbial Sciences, King's College London, London SE1 9RT, UK. [2] Centre for Cardiovascular Sciences, Queens Medical Research Institute, University of Edinburgh, Edinburgh EH16 4TJ, UK. [3] Biodesign Institute, Centre for Infectious Disease and Vaccinology, Arizona State University, Tempe, AZ 85287, USA. [4] TheraJect Inc, Fremont, CA 94538, USA. [5]Present address: Faculty of Biology, Medicine and Health, University of Manchester, Manchester M13 9PL, UK. Correspondence and requests for materials should be addressed to L.S.K. (email: linda.klavinskis@kcl.ac.uk)

The majority of pathogens, including HIV, influenza virus and Mycobacterium tuberculosis that contribute to the global burden of infectious disease initiate infection at the mucosal barrier tissues. At least two subsets of memory CD8[+] T cells patrol the mucosal tissues as a front line defence: a non-recirculating, tissue-resident memory ($T_{RM}$) population of CD8[+] T cells[1] and an effector memory ($T_{EM}$) population of CD8[+] T cells that circulate between the blood and non-lymphoid tissues[2]. These subsets of CD8[+] T cells are uniquely poised to provide an immediate response, within minutes to hours of pathogen detection at the barrier tissues and dictate the outcome of infection. In direct contrast, circulating central memory ($T_{CM}$) CD8[+] T cells require time to enter the tissues and respond to pathogen infection. Accordingly, intense interest has focused on immunisation strategies that promote recruitment of memory CD8[+] T-cell subsets to the mucosal epithelial tissues[1,3,4].

Traditionally, the generation of antigen-specific T cells that manifest defences at the epithelial barriers has been viewed as dependent on mucosal routes of vaccination[5–8], and priming by antigen-bearing dendritic cells (DCs) at the mucosa and associated draining lymphoid tissues[9,10]. Such DCs were found to imprint distinct homing molecules that help gut/lung-activated T cells to efficiently traffic to the anatomic site of pathogen infection or immunisation and clear the antigen[11–14]. Nonetheless, immunisation by systemic routes; intramuscular (IM), subcutaneous (S.C.) and intradermal (I.D.) with replication defective viral vectors, including adenovirus (Ad) and pox viruses can effectively elicit and recruit polyfunctional antigen-specific CD8[+] T cells to mucosal barrier tissues[15–18]. These responses can be long-lived and confer protection[17]. For example, scarification of the skin with vaccinia virus (VV) or modified VV Ankara protects from respiratory pathogen challenge in animal models[19,20]. Importantly in humans, skin scarification with VV remains a classic example of a successful vaccination strategy that confers protection against smallpox (variola virus) infection that is acquired by the aerosol route[21]. These reports would suggest that cutaneous antigen exposure does not restrict seeding of antigen experienced CD8[+] T cells (effector and memory) to the skin.

The mechanism/s that enable antigen-specific CD8[+] T cells to disseminate and acquire residence in epithelial barrier tissues remote from the site of immunisation are unclear. While it is well acknowledged that CD8[+] T cells are programmed to express adhesion and chemokine receptors specific to the site of immunisation, emphasising the role of tissue DCs in determining T-cell trafficking[22,23], alternative mechanisms for CD8[+] T-cell trafficking must exist in parallel to explain T-cell dissemination to remote, unrelated tissues. One paradigm states that a subset of early effector CD8[+] T cells may gain additional homing potential by trafficking through distant lymph node (LN) microenvironments, such that T cells originally activated in skin draining LNs may acquire gut-homing molecules by trafficking through mesenteric LNs[24]. Equally, there is evidence that the spleen engenders early CD8[+] T-cell effectors to acquire a gut-homing programme through expressed α4β7, though migration to non-intestinal tissues was not addressed[25]. It is also known that inflammatory signals provided by topical chemokine application to the vagina can recruit activated CD8[+] T cells from the systemic compartment to that tissue and establish a tissue-resident memory pool by a 'prime-pull' strategy[26]. However, the capacity of (i) early spleen-derived CD8[+] T-cell effectors to migrate to epithelial tissues despite their not being the target of infection and (ii) the potential crosstalk of innate effectors (including macrophages, innate lymphoid cells and unconventional T cells) within the epithelia that may provide signals to recruit CD8[+] T cells remains unanswered. Given the importance of immune surveillance at the epithelial barrier tissues and to reconcile seemingly disparate reports, we set out to gain further insights into the mechanism by which skin immunisation generates CD8[+] T cells that populate the mucosal epithelia.

Here, we report using a replication-defective Ad5 vector encoding HIV-1 CN54 gag (Ad-CN54 gag) as a candidate vaccine antigen, that skin immunisation generated CD8[+] T cells that uniformly expressed the chemokine receptor CXCR3 and conferred protection in the female reproductive mucosa against a VV challenge encoding the cognate antigen. We discovered that following skin immunisation, a low but consistent copy number of vector DNA biodistributed to the female reproductive tract (FRT). Importantly, while this low dose of biodistributed vector was insufficient to prime CD8[+] T cells locally in the FRT, it was sufficient to recruit skin primed CXCR3[+]CD8[+] T-cells independent of antigen by CXCL9 (the cognate chemokine ligand for CXCR3) produced by CD11b[+]Ly6C[+] monocytes and also by CD11b[+] DC. Cell intrinsic interferon gamma (IFNγ) produced by group 1 innate lymphoid cells (ILC1) in the FRT was essential to licence CXCL9 expression and recruit CD8[+] T cells to the tissue. These results reveal a previously unappreciated role for ILC1 to mobilise effector CD8[+] T cells to peripheral epithelial tissues and provide a memory pool of tissue-resident CD8[+] T cells to protect the host against infection.

## Results

**Skin immunisation elicits high-frequency CD8 T -cells in FRT.** We first investigated the capacity of skin immunisation to generate antigen-specific CD8[+] T cells that localise in the female reproductive mucosa. As a model system, B6 mice were immunised with a relevant vaccine target, a synthetic HIV-1 gag CN54 gene (codon optimised for mammalian codon use) and expressed from a replication defective Ad5 vector (Ad-CN54 gag) described previously[27]. Immunisation was performed by intradermal injection (ID) and by a promising microneedle array (MA) platform progressing through the clinic[27], whose immunising properties we reported are dependent on dermal DCs[27]. At $1 \times 10^9$ viral particles (vp) Ad-CN54 gag, both skin immunisation routes elicited as expected high frequency CD8[+] T cells tracked by tetramer (Tet) to the immunodominant D[b]-restricted HIV-1 CN54 gag$_{308-318}$ epitope (D[b]/CN54 gag)[17] in blood, spleen and inguinal LN (Fig. 1a–d). Remarkably, both cutaneous routes of immunisation-elicited high frequency D[b]/CN54 gag Tet[+] CD8[+] T cells that populated the FRT (Fig. 1d). The lower Ad vector dose ($1 \times 10^7$ vp) applied to the skin, elicited a markedly lower frequency of D[b]/CN54 gag Tet[+] CD8[+] T cells recruited to the FRT (Supplementary Fig. 1) consistent with previous Ad5 vaccination studies that report a vector dose-dependence for priming systemic antigen-specific CD8 T cells[27,28].

To exclude the possibility that proximity of the immunisation site (back skin) influenced accumulation of CD8[+] T cells in the female FRT, mice were immunised to the dorsal surface of the ear (Fig. 1e). Irrespective of the site of skin immunisation (ear or back), the frequency of activated CD11a[hi]D[b]/CN54gag Tet[+] CD8[+] T cells detected in the FRT at day 14-post immunisation was statistically indistinguishable (Fig. 1f, g). Having established that skin immunisation has the capacity to recruit antigen-specific CD8[+] T cells to the FRT, we examined if this observation might be extended to IM immunisation, a route commonly used in human vaccination. There was a lower but not-significant difference in frequency of CN54gag Tet[+] CD8[+] T cells recruited to the FTR after IM immunisation when cross-compared with skin immunisation (Supplementary Fig 2). Together these data demonstrated that antigen-specific CD8[+] T cells primed by skin (and also IM) immunisation localise not only in lymphoid tissue

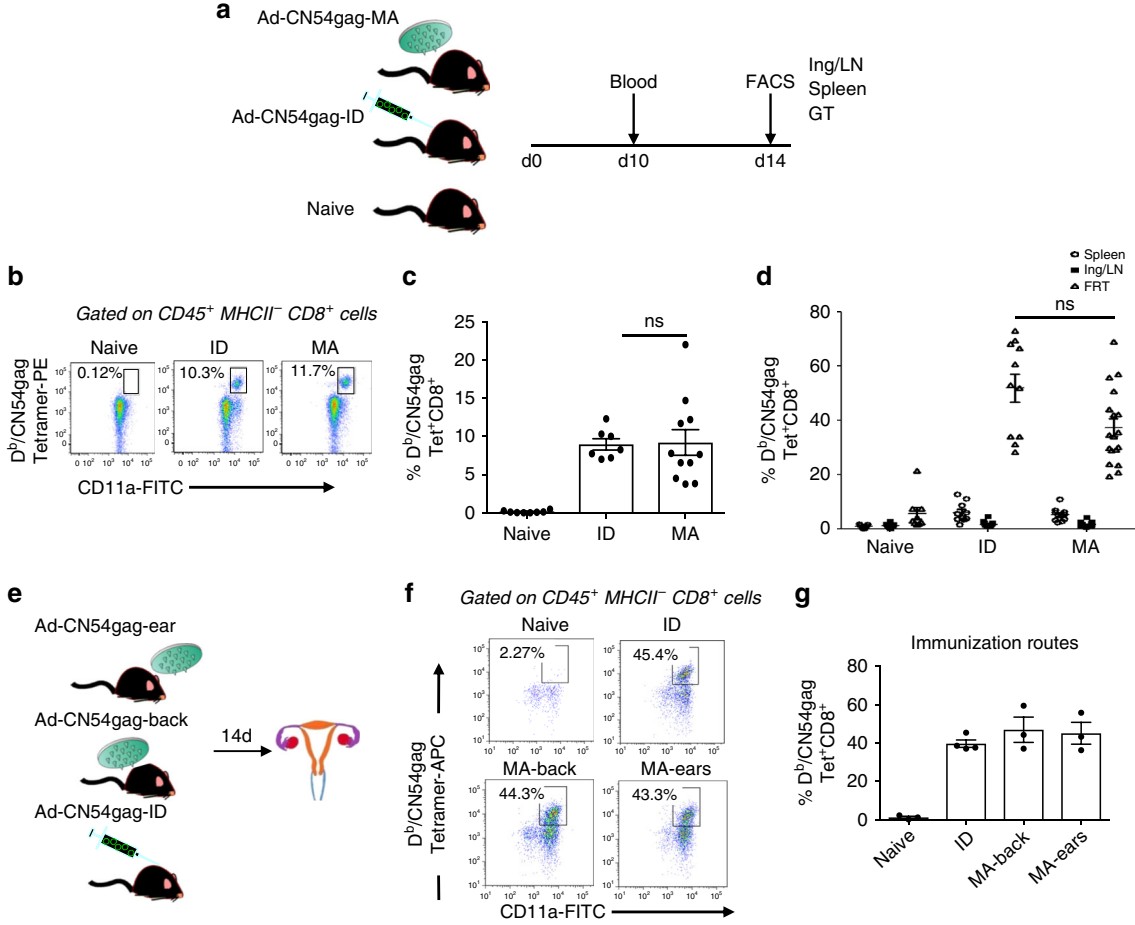

**Fig. 1** Skin immunisation irrespective of anatomic site recruits CD8+ T cells to the FRT. **a** Experimental design. **b–d** Flow-cytometry plots and graphs showing frequencies of CD8+ $D^b$/CN54gag tet+ cells in blood (day 10) and indicated tissues (spleen, inguinal LN and FRT) on day 14 after ID or MA (back skin) immunisation with Ad-CN54gag. Data are from four independent experiments ($n = 7–18$ mice per group). **e** Experimental design. Mice were immunised by MA either to the dorsal surface of the ear or to back skin or ID. **f, g** Flow-cytometry plots and graph showing frequencies of CD8+ $D^b$/CN54gag Tetramer+ cells in FRT (day 14). Data are representative of two independent experiments ($n = 3–4$ mice /group). **d, g** $p > 0.05$ (ns) by one-way ANOVA, error bars show s.e.m

draining the site of antigen encounter but also in non-lymphoid tissue such as the FRT by a mechanism independent of the anatomic site of immunisation or the mode of antigen delivery.

**Skin primed FRT CD8 T- cells protect against VV challenge.** Given that antigen experienced CD8+ αβ+ T cells may express regulatory or cytotoxic effector functions in mucosal tissues[29–32] that are acquired during priming[33], we determined if CD8+ T cells localised in the FRT after skin immunisation expressed cytolytic anti-viral activity. To that end, mice primed with Ad-CN54 gag by ID injection or by MA immunisation were challenged by injection into the wall of the vagina on day 14 with carboxyfluorescein succinimidyl ester (CFSE)-labelled syngeneic targets pulsed with peptide to the immunodominant Db-restricted HIV-1 CN54 gag epitope[34] and with control targets. Cytotoxicity of CFSE-labelled donor cells isolated from the vagina was analysed after 18 h by flow cytometry (Fig. 2a). Robust antigen-specific in vivo killing was observed in the vagina of mice 14 days after skin immunisation (either ID or by MA) (Fig. 2b, c). Furthermore, flow cytometry of cells isolated from the FRT tissue of immunised mice, when re-stimulated in vitro with peptide to the immunodominant Db-restricted CN54 gag epitope demonstrated a substantial population of antigen-specific CD45+ MHCII− CD8+ cells expressed biomarkers of anti-viral CD8

effector T cells, IFNγ and/or Granzyme B (Fig. 2d, e). When immunised or naive mice were challenged intravaginally 4 weeks post ID or MA immunisation with $5 \times 10^7$ plaque-forming units (pfu) recombinant VV expressing the cognate HIV-1 gag CN54 antigen (Fig. 2f), we found a significant decrease in viral titres within the FRT of the immunised mice when cross-compared with the naive group ($P < 0.001$, one-way ANOVA) on day 6 post challenge (Fig. 2g). These data indicate that immunisation through the skin has the capacity to induce a robust and protective CD8+ T-cell response in the FRT.

**Biodistributed Ad5 is insufficient to prime FRT CD8 T- cells.** To determine the mechanism by which antigen-specific CD8+ T cells seed the FRT following skin immunisation, we first enumerated Ad5 vector genomes by quantitative PCR in several tissues at an early time point after immunisation to address if virus biodistributed from the skin to the FRT and if that was sufficient to locally prime CD8+T cells within the female reproductive tissue. At 24 h post MA immunisation with Ad-CN54gag ($1 \times 10^9$ vp), a low but reproducible copy number of Ad genomes were detected in the FRT tissue, liver, lung and spleen (Fig. 3a). While the genome content in these tissues was significantly lower than that detected in the skin ($p < 0.001$) or skin-draining LNs ($p < 0.05$) by one-way ANOVA, it was consistently and markedly

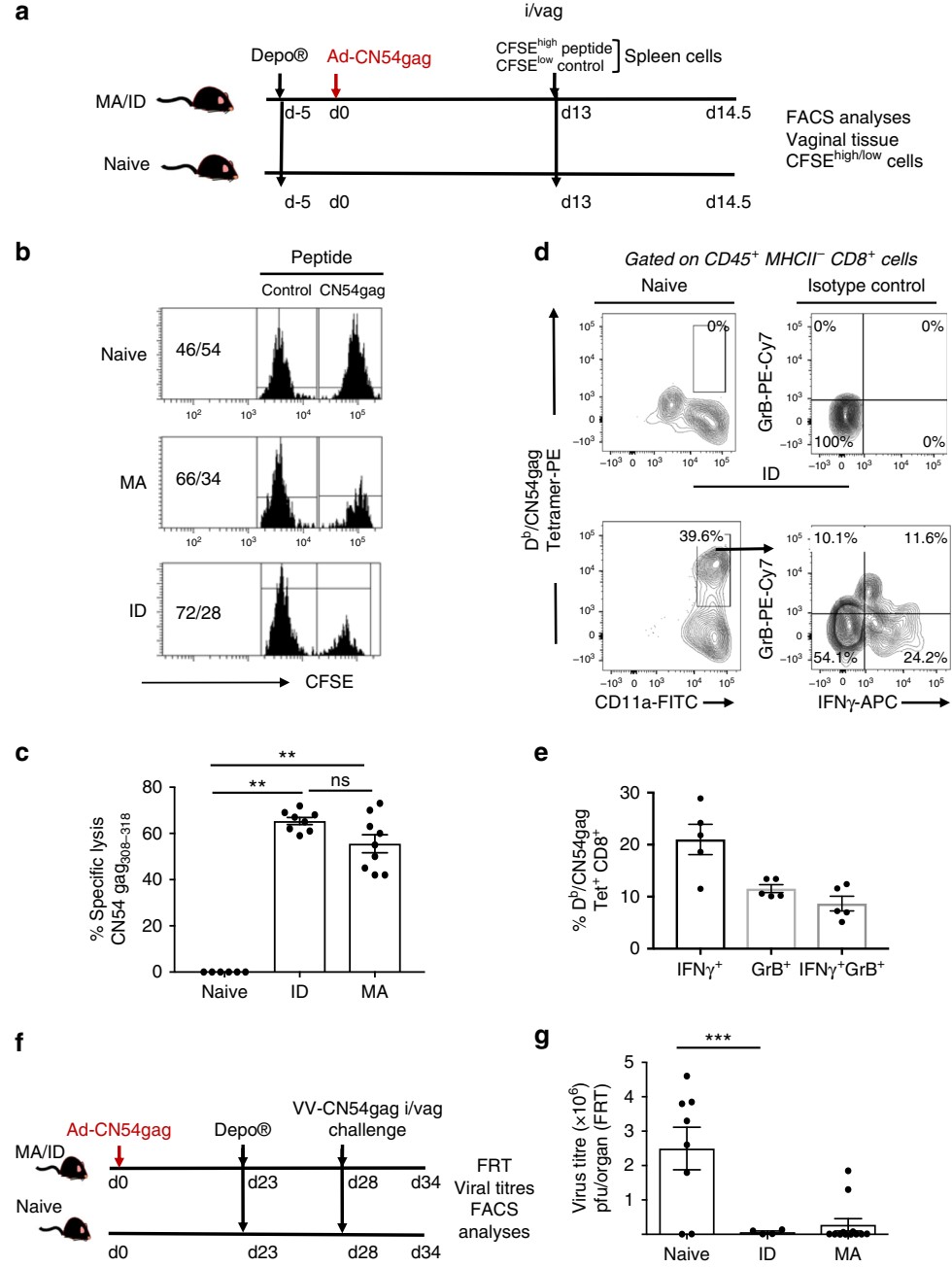

**Fig. 2** FRT CD8[+] T cells induced by skin immunisation are cytolytic and protect against vaginal virus challenge. **a** Experimental design. Syngeneic donor spleen cells pulsed with peptide to the immunodominant Db/HIV-1 CN54 gag epitope (CFSE[high]) and control non-pulsed (CFSE[low]) syngeneic donor spleen cells (10[6] at 1:1 ratio) were injected into the vaginal mucosa of Depo-Provera treated, Ad-CN54 gag immunised /naive mice. After 18 h, CFSE-labelled donor cells isolated from the vaginal mucosa analysed by flow cytometry. **b** Representative flow-cytometry CFSE profiles. **c** Percentage-specific lysis of HIV-1 CN54 gag peptide-coated donor cells in vagina of recipient mice. Data are from two independent experiments ($n = 6$–9 mice /group), **$p < 0.01$ by one-way ANOVA, error bars show s.e.m. **d**, **e** Representative flow-cytometry profiles and summary graph of IFNγ and/or Granzyme B expressed by CD8[+] T cells isolated from FRT tissue 14 days after Ad-CN54 gag skin immunisation and stimulated in vitro with peptide specific to the immunodominant Db/CN54 gag epitope. Data represent $n = 5$ mice/group. **f** Experimental design for testing host protection from vaginal infection with rVV-CN54gag ($5 \times 10^7$ pfu) conferred by skin immunisation (ID or MA) with Ad-CN54gag. **g** Virus (VV) titres in the vaginal mucosa were measured by plaque assay on day 6 post virus challenge. Data are representative of two independent experiments with a total of 4–12 mice per group. ***$p < 0.001$ by one-way ANOVA, error bars show s.e.m

greater than the Ad genome content (per 100 ng of DNA) in blood (Fig. 3a, b), indicating that virus biodistributed from the skin to peripheral non-lymphoid tissues. The capacity of this low dose of biodistributed virus particles to prime CD8[+] T cells locally in the vaginal mucosa was assessed by intravaginal injection of naive mice with $3.5 \times 10^2$ vp Ad-CN54 gag that was equivalent to

the Ad5 genome content detected in the FRT after skin immunisation. Tracking by tetramer did not reveal D[b]/CN54 gag Tet[+] CD8[+] T cells in the iliac LNs (Fig. 3c) or FRT at day 7 or 14 after intravaginal immunisation with $3.5 \times 10^2$ vp Ad -CN54 gag (Fig. 3d; Supplementary Fig. 3). In contrast, the skin-priming dose ($1 \times 10^9$ vp) when injected intravaginally elicited a high

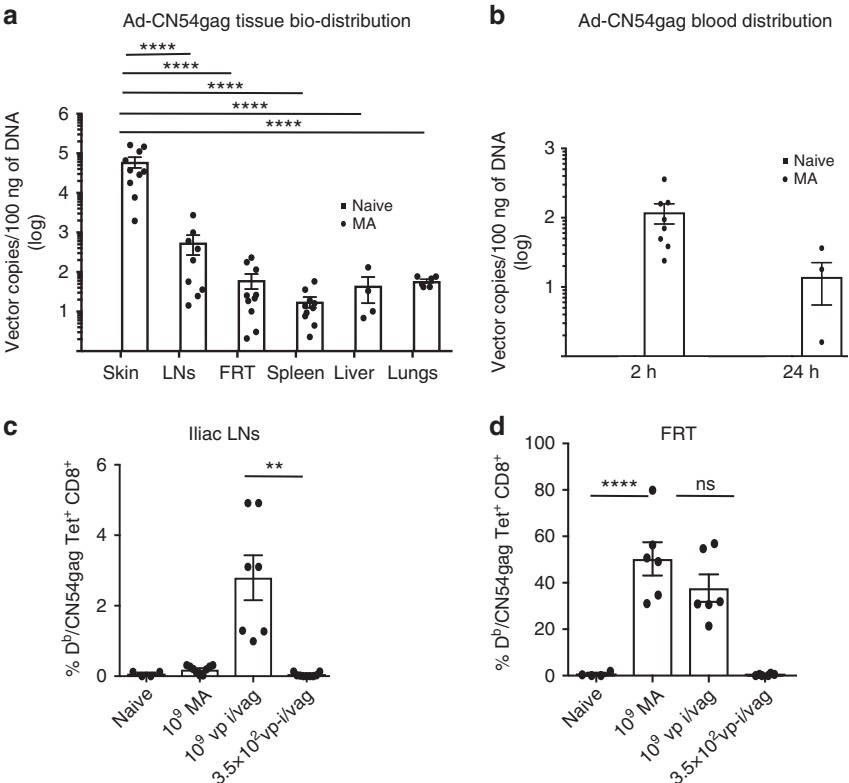

**Fig. 3** Biodistributed rAd5 vaccine vector does not prime CD8[+] T cells in FRT or iliac LNs. **a** Quantification of Ad vector genomes by qPCR sequestered in indicated organs at 24 h post MA immunisation with Ad-CN54 gag (1×10$^9$ vp). **b** Ad vector genomes quantified in whole blood at 2 and 24 h post skin (MA) immunisation with Ad-CN54 gag (1×10$^9$ vp) (**a**, **b**). Data are from two independent experiments (n = 4–10 per group). **c**, **d** Frequencies D$^b$/CN54 gag tet $^+$ CD8$^+$T cells in FRT tissue and iliac LNs after skin (MA) or intravaginal immunisation with Ad-CN54 (1×10$^9$ vp) or with the vector dose (3.5 × 10$^2$ vp) equivalent to the copy number detected by qPCR in FRT after skin immunisation. Data (n = 4–9 per group) are from three independent experiments. $P$>0.05 (ns), **$p$ < 0.01, ***$p$ < 0.001 by one-way ANOVA, error bars show s.e.m

frequency of CD11a$^{hi}$D$^b$/CN54 gag Tet$^+$ CD8$^+$ T cells in the FRT (Fig. 3d; Supplementary Fig. 3a, b) that was not statistically different from that elicited in the FRT after skin immunisation by MA ($p$ = 0.23 by one-way ANOVA). Therefore, the Ad-CN54 gag virus dose biodistributed from the skin to the FRT was below the threshold required to prime antigen-specific CD8$^+$T cells locally in the vagina and iliac LNs.

**FRT CD8 T-cell recruitment requires CXCR3 independent of Ag.** We next addressed if effector CD8$^+$ T cells from the systemic compartment gained the ability to migrate to the FRT after skin immunisation and if specific cues drove their migration. To that end, D$^b$/CN54 gag Tet$^+$ CD8$^+$ T cells in blood and FRT at day 7 after skin immunisation were screened by flow cytometry against a chemokine receptor and integrin panel of acknowledged homing molecules. Of note, chemokine receptors associated with inflammation (CCR1, CXCR6) were expressed at high frequency (Supplementary Fig. 4) with CXCR3 universally expressed (>98%) on D$^b$/CN54 gag Tet$^+$ CD8$^+$ T cells in the blood and FRT at day 14 after skin (ID or MA) immunisation (Fig. 4a). These observations were not restricted to the genital mucosal tissues. High frequency activated D$^b$/CN54 gag Tet$^+$ CD8$^+$ T cells were detected in the lung after ID/ MA immunisation (Supplementary Fig. 5a) and the majority expressed CXCR3 (Supplementary Fig. 5b). Likewise, the CD8$^+$ Tet$^-$ D$^b$/CN54gag$_{308–318}$ population from the FRT of immunised but not naive mice was enriched for CXCR3$^+$ expression ($p$ < 0.05 by one-way ANOVA) (Supplementary Fig 6). This may indicate CXCR3$^+$CD8$^+$ T-cell effectors that are primed against additional

TCR specificities including the vector and the gag protein are recruited to the FRT.

To test if CXCR3 expression was necessary to permit spleen-derived CD8$^+$ T-cell effectors to migrate to the FRT after immunisation, we used a transfer system of congenic Rag$^{−/−}$ CD45.1 T-cell receptor transgenic OT-I CD8$^+$ T cells (CD45.1 OT-I) specific for the ovalbumin (OVA) peptide 257–264 to track migration in groups of wild-type (WT) recipients immunised in the skin (1 × 10$^9$ vp Ad-OVA) or by intravaginal injection with the biodistributed virus dose (3.5 × 10$^2$ vp Ad-OVA / Ad-CN54 gag). To generate effector CD8$^+$ T cells, 2 × 10$^5$ naive CD45.1 OT-I cells were transferred into WT mice. Six days after Ad-OVA skin immunisation, some were treated with anti-CXCR3 Ab while others were treated with isotype control Ab. On day 7, effector CD45.1 OT-I cells (2 × 10$^6$) isolated from anti-CXCR3 Ab-treated or isotype control Ab-treated mice were transferred into secondary recipients (treated with anti-CXCR3 Ab or isotype control Ab), which had been immunised by MA or by intravaginal injection 3.5 days earlier (Fig. 4b). Flow cytometry confirmed that surface CXCR3 was blocked on cells from anti-CXCR3-treated mice (Supplementary Fig. 7). As a control for antigen specificity, some secondary recipients were intravaginally injected with Ad-CN54 gag or PBS 3.5 days prior to effector OT-I cells transfer (Fig. 4b). After 1.5 days, CD45.1 OT-I cells were quantified from recipient tissues. The number of CD45.1 OT-I effectors recovered from the blood and spleen was similar in naive and immunised mice, irrespective if the transferred cells expressed CXCR3 or were blocked for CXCR3 (Fig. 4c, d). However, as expected transferred CXCR3$^+$CD45.1 OT-I cells

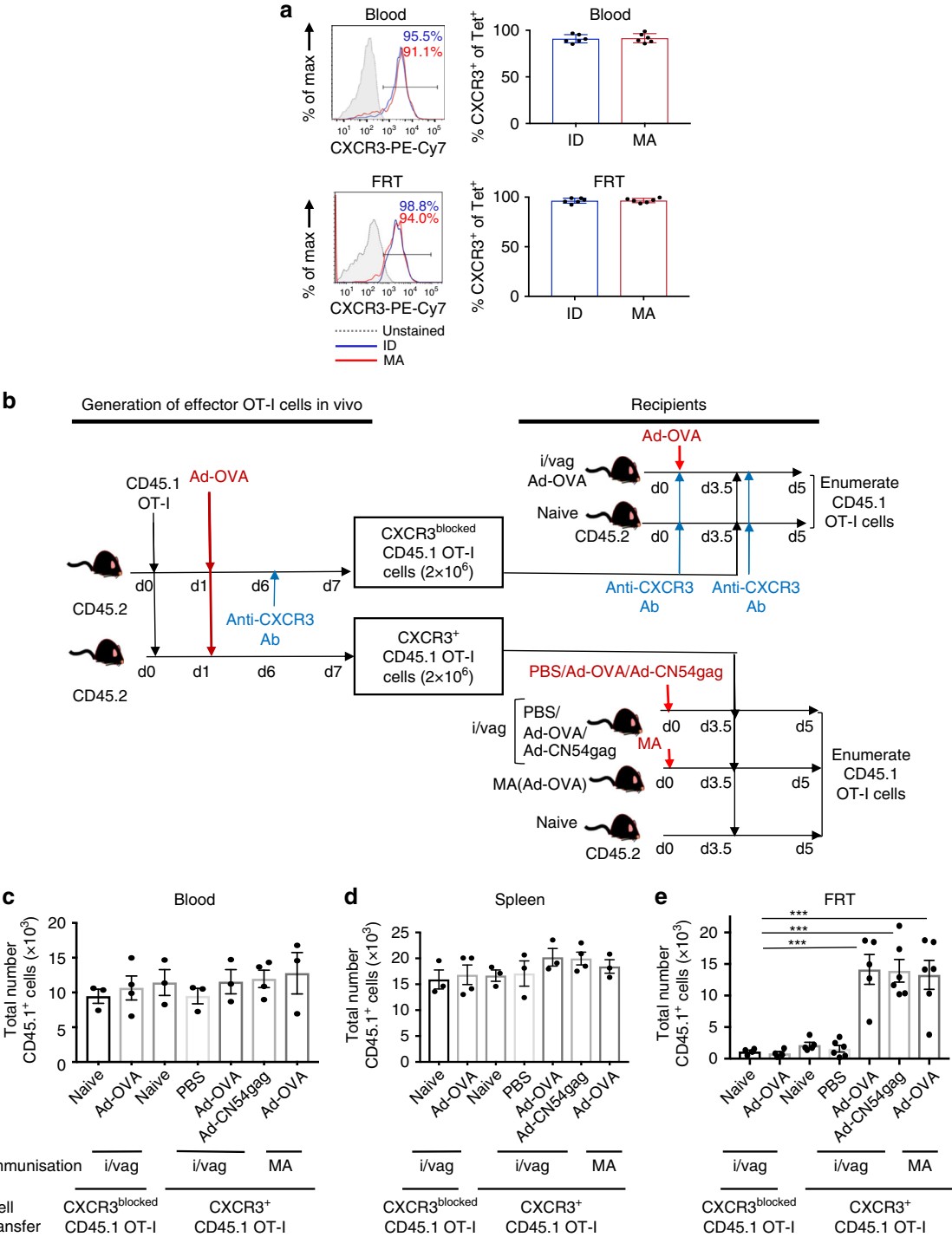

**Fig. 4** CXCR3 is essential for recruitment of CD8+ T cells to FRT, independent of antigen specificity. **a** Representative flow-cytometry profiles and graphs showing frequency of CXCR3+ D^b/CN54 gag tet+ cells in blood (upper panels) and FRT (lower panels) 14 days after skin immunisation; ID (blue histograms) or MA (red histograms); grey-filled histograms represent unstained control, all gated from a CD45+CD4−MHCII−CD8+ population. Bar graphs summarise data from two independent experiments ($n = 6$ for experimental groups). **b** Experimental design. Congenic CD45.1 OT-I cells (2 × 10^5) were transferred into WT mice. Five days after Ad-OVA skin immunisation, some were treated with anti-CXCR3 Ab. On day 7, effector OT-I cells (2 × 10^6) isolated from these mice were transferred into secondary hosts (some treated with anti-CXCR3 Ab) infected with either Ad-OVA via MA or intravaginal injection 3.5 days earlier. Some secondary recipients were intravaginally injected with Ad-CN54 gag or PBS 3.5 days prior to effector OT-I cell transfer. **c**–**e** Numbers of CD45.1+OT-I cells in the blood, spleen and FRT tissues assessed 5 days post immunisation. The data are pooled from two independent experiments ($n = 4$–6 per experimental group). ***$p < 0.001$ by one-way ANOVA, error bars show s.e.m

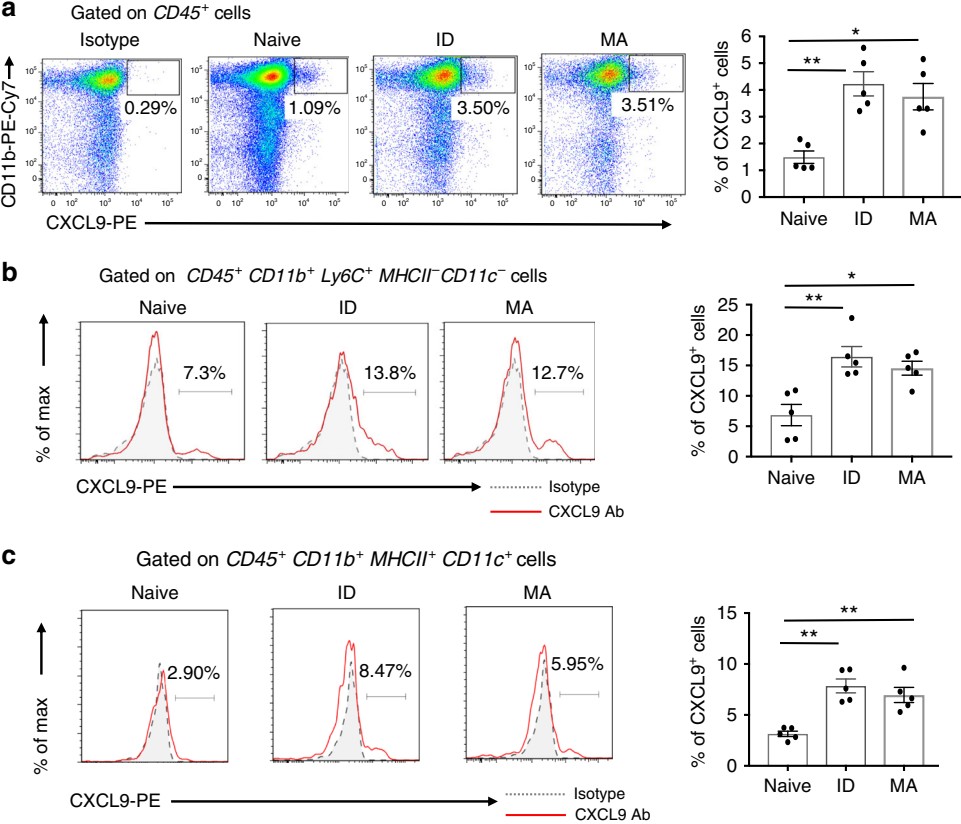

**Fig. 5** Monocytes infiltrating the FRT express the CXCR3 ligand CXCL9 after skin immunisation. **a** Representative flow-cytometry plots and summary bar graph of CXCL9 expressed on CD45+ CD11b+ cells isolated from FRT tissue at day 7 after skin immunisation with Ad-CN54 gag via ID injection or MA application or left naive. **b**, **c** Representative histograms and bar graphs summarise CXCL9 expression among (**b**) a gated CD45+Ly6C+CD11b+MHCII−CD11c− monocyte population and (**c**) a gated CD45+Ly6C−CD11b+MHCII+CD11c+ dendritic cell population. Data are representative of three independent experiments (n = 5 per experimental group). *p < 0.05, **p < 0.01 by one-way ANOVA, error bars show s.e.m

were excluded from the FRT of naive recipients. In contrast, a significant number of transferred CXCR3+CD45.1 OT-I cells were recovered from the FRT, both of mice immunised with Ad-OVA in the skin and from recipients immunised intravaginally with the low virus dose that biodistributed from the skin to FRT (Fig. 4e). Moreover, migration of transferred CD45.1 OT-I cells to the FRT that was independent of the antigen encoded by the intravaginally injected Ad vector, but was absolutely dependant on CXCR3 expression of the transferred cells, as treatment with anti-CXCR3 Ab inhibited CD45.1 OT-I cells homing to the FRT (Fig. 4e). These data suggest that the low level of virus that biodistributes from the skin to the FRT is sufficient to recruit systemic CD8+ T-cell effectors to the FRT and that migration was dependant on expressed CXCR3, but independent of antigen specificity.

**Skin immunisation upregulates CXCL9 on FRT myeloid cells.** To test the hypothesis that innate cells in peripheral non-lymphoid tissues remote from the site of immunisation played a role in CD8+ T-cell recruitment to that tissue, we examined myeloid populations for expression of CXCL9, an effector chemokine and cognate ligand of CXCR3[35] in the FRT after skin immunisation. We found a significant increase in FRT CD45+ CD11b+ cells that expressed CXCL9 from skin immunised mice, both ID and MA when compared with naive (Fig. 5a). Likewise in the lung, there was a marked increase in CXCR9+CD45+CD11b+ cells after skin immunisation (Supplementary Fig. 8). Deeper phenotypic analysis of CD11b+ cells revealed a distinct and significant increase in the frequency of cells that expressed CXCL9

within the CD45+Ly6Chi CD11b+ MHCII− CD11c− population, hereafter referred to as 'Ly6Chi monocytes' (Fig. 5b) and within the CD45+CD11b+ MHCII+ CD11c+ population referred to as 'dendritic cells' (Fig. 5c) in the FRT at day 7 and for at least 14 days post skin immunisation (Supplementary Fig 9a, b). Collectively, the data suggest that skin immunisation with an Ad-vectored vaccine engenders expression of CXCL9 by CD11b+ subsets in peripheral non-lymphoid barrier tissues.

**FRT monocytes and ILC1 accumulate after skin immunisation.** Our observation that monocytes within the FRT expressed CXCL9, an IFNγ-inducible chemokine in the context of skin immunisation, raised the possibility that production of CXCL9 may be enforced by innate lymphoid cells, such as classical natural killer cells reported capable of producing IFNγ[36,37]. There are precedents to support this idea as adenovirus is a strong inducer of NK cell activation[28,38,39] and NK cell-derived IFNγ can prime monocyte functions[40]. To that end, we assessed the kinetics of monocyte and 'NK-like' ILC1 accumulation in the blood and FRT after skin (MA) immunisation with Ad-CN54 gag. As expected, NK-like ILC1 (defined as CD3− NK1.1+ and referred to hereafter as ILC1) and Ly6Chi monocytes rapidly accumulated in the blood within 3 h of skin immunisation, peaked after 6 h and returned to baseline at 24 h, with a second influx (increase) in blood Ly6Chi monocytes at 48 h after immunisation (Fig. 6a; Supplementary Fig. 10). Contrastingly, a peak in frequency of ILC1 within the FRT was evident at 24 h that represented a 2.2-fold increase compared with naive mice (Fig. 6b), with CD69 (an early activation and tissue retention

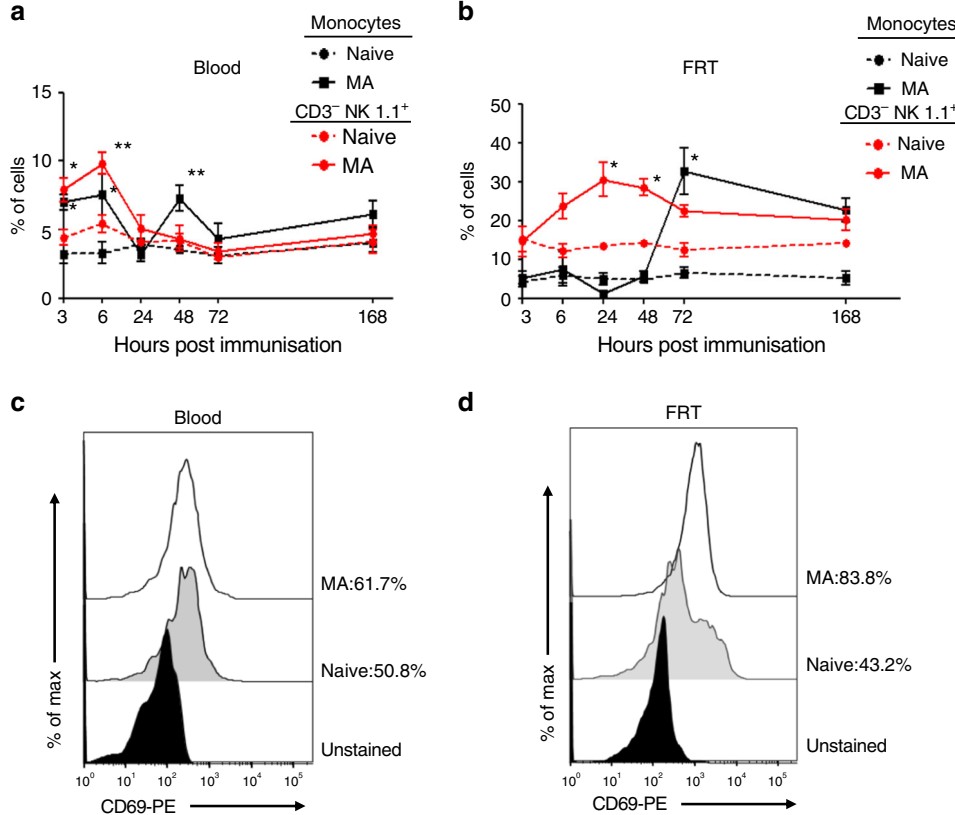

**Fig. 6** Skin immunisation promotes the early recruitment of ILC1 and monocytes to the FRT. **a, b** Frequency of ILC1 (CD45+NK1.1+ CD3−) and monocytes (CD45+Ly6C+CD11b+MHCII−) in peripheral blood (**a**) and FRT tissue (**b**) at the indicated time points from naive or Ad-CN54 gag skin MA immunised mice (**c, d**). Representative flow-cytometry profile of CD69 expressed on ILC1 isolated from peripheral blood (**c**) or FRT tissue (**d**), 24 h after Ad-CN54 gag immunisation or from naive controls. Data are from three independent experiments (n = 3–10 per group). *p < 0.05, **p < 0.01 by one-way ANOVA, error bars show s.e.m

marker) highly expressed on the majority of ILC1 (Fig. 6b). A significant influx of Ly6C$^{hi}$ monocytes within the FRT was not apparent until 72 h post skin immunisation (Fig. 6b), but remained elevated for at least 14 days (Supplementary Fig. 11). Thus, after skin immunisation with an Ad5-vectored vaccine, there is a marked accumulation of CD69+ ILC1 in the FRT that have the potential to engage in crosstalk with recruited Ly6C$^{hi}$ monocytes.

**Biodistributed Ad5 sufficient to recruit monocytes and ILC1.** Next, we addressed if biodistribution of virus from the skin to peripheral tissues in the context of skin immunisation was sufficient to induce the observed increase in ILC1 and Ly6C$^{hi}$ monocytes within the FRT. Mice were administered Ad-CN54 gag, either $1 \times 10^9$ vp by MA to the skin or $3.5 \times 10^2$ vp (equivalent to the biodistributed virus dose) by injection into the wall of the vagina (Fig. 7a). At 24 h post virus immunisation, a substantial increase in frequency and the total number of ILC1 was detected in the FRT of immunised relative to naive mice (p < 0.001 and p < 0.01, respectively, by one-way ANOVA) that was not significantly different if the low (biodistributed equivalent) virus dose was injected intravaginally as opposed to skin priming by MA (Fig. 7c; Supplementary Fig. 7). Likewise, at 72 h post immunisation, the low virus dose injected intravaginally led to a significant increase in frequency and the total number of Ly6C$^{hi}$ monocytes within the FRT relative to naive mice (p < 0.01, Fig. 7e) that was not significantly different (p>0.05) from mice immunised by MA in the skin (Fig. 7e; Supplementary Fig. 12) analysed by one-way ANOVA. Of note, the stress response from

injection into the vaginal wall (with PBS) did not account for the increase in FRT ILC1 or Ly6C$^{hi}$ monocytes associated with the intravaginally injected low virus dose (Fig. 7e; Supplementary Fig. 12). These data strongly suggest that virus biodistributed from the skin to the FRT after skin immunisation is sufficient to modify the peripheral tissue environment such that this resulted in recruitment of ILC1 and inflammatory monocytes to the FRT tissue.

**CD8 T-cell recruitment to FRT requires ILC1 intrinsic IFNγ.** To better understand the role of innate lymphoid-like cells in recruitment of CD8+ T cells to the FRT, we treated mice with an anti-NK1.1-depleting antibody 1 day prior to and at 3-day intervals following skin immunisation with Ad-CN54 gag (Fig. 8a). This treatment resulted in nearly complete ablation of the CD45+ CD3− ILC1 population in the FRT and CD3−CD19− ILC1 cells in the blood (Supplementary Fig. 13). Strikingly, we observed that anti-NK1.1 antibody treatment was associated with a significant decrease in the frequency of D$^b$/CN54 gag Tet+ CD8+ T cells in the FRT when compared with the control treated group (p < 0.01, by one-way ANOVA, Fig. 8c). Conversely, there was no such substantial decrease in D$^b$/CN54 gag Tet+ CD8+ T cells in the blood after anti-NK1.1 antibody treatment, Fig. 8b). Moreover, expression of CXCR3+ remained unchanged on D$^b$/CN54 gag Tet+ CD8+ T cells in the blood or FRT after anti-NK1.1 antibody treatment (Fig. 8d, e). Thus CD8+ T-cell recruitment to the FRT after skin immunisation showed a strong dependence on ILC1 within the FRT, despite this tissue not being the site of immunisation.

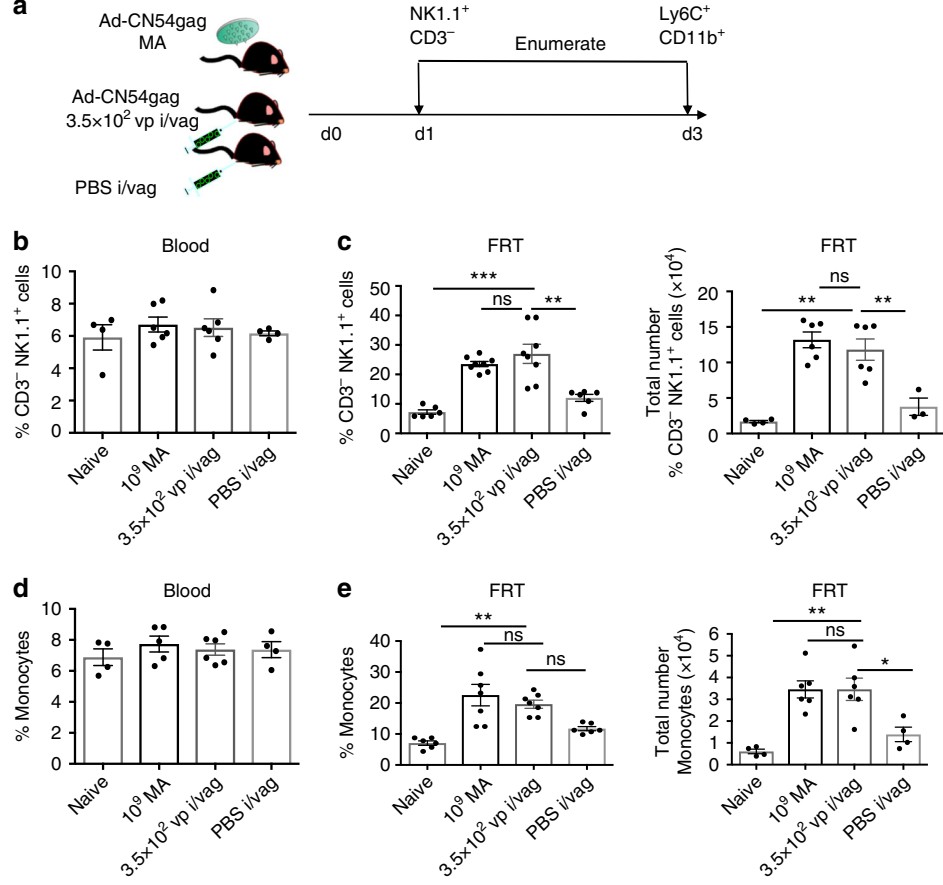

**Fig. 7** Ad vector biodistributed after skin immunisation recruits group 1 ILCs and monocytes into the FRT. **a** Experimental design. Mice were immunised with Ad-CN54 gag by MA delivery to the skin ($1 \times 10^9$ vp) or by injection to the vaginal mucosa (i/vag) with $3.5 \times 10^2$ vp (equivalent to the biodistributed Ad vector dose). Control mice were injected PBS (i/vag). After 1 day, NK1.1+ CD3− (ILC1) and after 3 days (monocytes) were enumerated from peripheral blood and FRT tissues. **b** Frequency of NK1.1+ CD3− ILC1 in peripheral blood. **c** Frequency (left panel) and absolute number (right panel) of NK1.1+ CD3− ILC1 in FRT. **d** Frequency of monocytes in peripheral blood. **e** Frequency (left panel) and absolute number (right panel) of monocytes cells in FRT. Data are from two independent experiments ($n = 4$–7 per experimental group). $p > 0.05$ (ns), $*p < 0.05$, $**p < 0.01$, $***p < 0.001$ by one-way ANOVA, error bars show s.e.m

Since accumulation of ILC1 preceded peak expansion of inflammatory monocytes in the FRT (Fig. 6b) and expression of CXCR3 was essential for CD8+ T-cell recruitment (Fig. 4), we reasoned that ILC1 might be crucial to programme expression of CXCL9 by Ly6C[hi] monocytes and enable CXCR3-driven CD8+ T-cell recruitment to the FRT. Treatment with anti-NK1.1 significantly decreased the frequency of CXCL9+ Ly6C[hi] monocytes in the FRT at 72 h after skin immunisation compared with immunised control treated mice (Fig. 8f, g) consistent with the hypothesis that innate lymphoid cells play a key role in priming CXCL9 expression by FRT inflammatory monocytes in the context of skin immunisation. That IFNγ might be essential in priming CXCL9 by Ly6C[hi] monocytes in this system was addressed by injecting wild-type and RAG-1-deficient ($Rag1^{-/-}$) mice that lack conventional T cells, NKT cells and B cells[41] with anti-IFNγ on the day prior to and daily after skin immunisation with Ad-CN54 gag (Fig. 8h). Anti-IFNγ significantly impaired the increase in frequency of CXCL9+ Ly6C[hi] monocytes within the FRT associated with skin immunisation (Fig. 8i, j) both in wild-type and $Rag1^{-/-}$ mice, suggesting an innate cell intrinsic IFNγ source was necessary to induce CXCL9. Consistent with that concept, the CXCL9 response of FRT Ly6C[hi] monocytes following skin immunisation with Ad-CN54 gag was unaltered in $Rag2^{-/-}\gamma c$-deficient ($Rag2^{-/-}$ X IL-2Rγ[null]) mice (that lack innate lymphoid cells) when cross-compared with the response of naïve $Rag2^{-/-}\gamma c$-deficient and in

contrast to wild-type immunised mice (Fig. 8j). Taken together, these data are consistent with the conclusion that innate lymphoid cell intrinsic IFNγ in the local environment of the FRT is necessary and sufficient to promote CXCL9 production by Ly6C[hi] monocytes and licence recruitment of CXCR3+ CD8+ T cells to the FRT following skin immunisation.

## Discussion

Innate lymphoid cells, emphasised by the group 1 subset are recognised to play an important role in protecting the epithelial barriers against intracellular infection[37,42,43]. This has been attributed to their capacity to rapidly produce IFNγ in response to local pro-inflammatory cytokines[40,44]. In this way, they directly restrict early pathogen replication at the initial site of infection[42–46]. Until now, it has been unappreciated that signals communicated by innate lymphoid cells can also permit CD8+ T cells primed after skin immunisation (or infection) to seed distal epithelial tissues, despite these tissues not being the target of infection. In this way, innate lymphoid cells can indirectly provide protection at the epithelia from secondary pathogen spread or future infection via promoting T-cell recruitment. Here, we elucidate how this occurs using an Ad vaccine vector. After skin immunisation, vaccine vector biodistributes at very low levels to peripheral tissues, including the FRT that is sufficient to mobilise

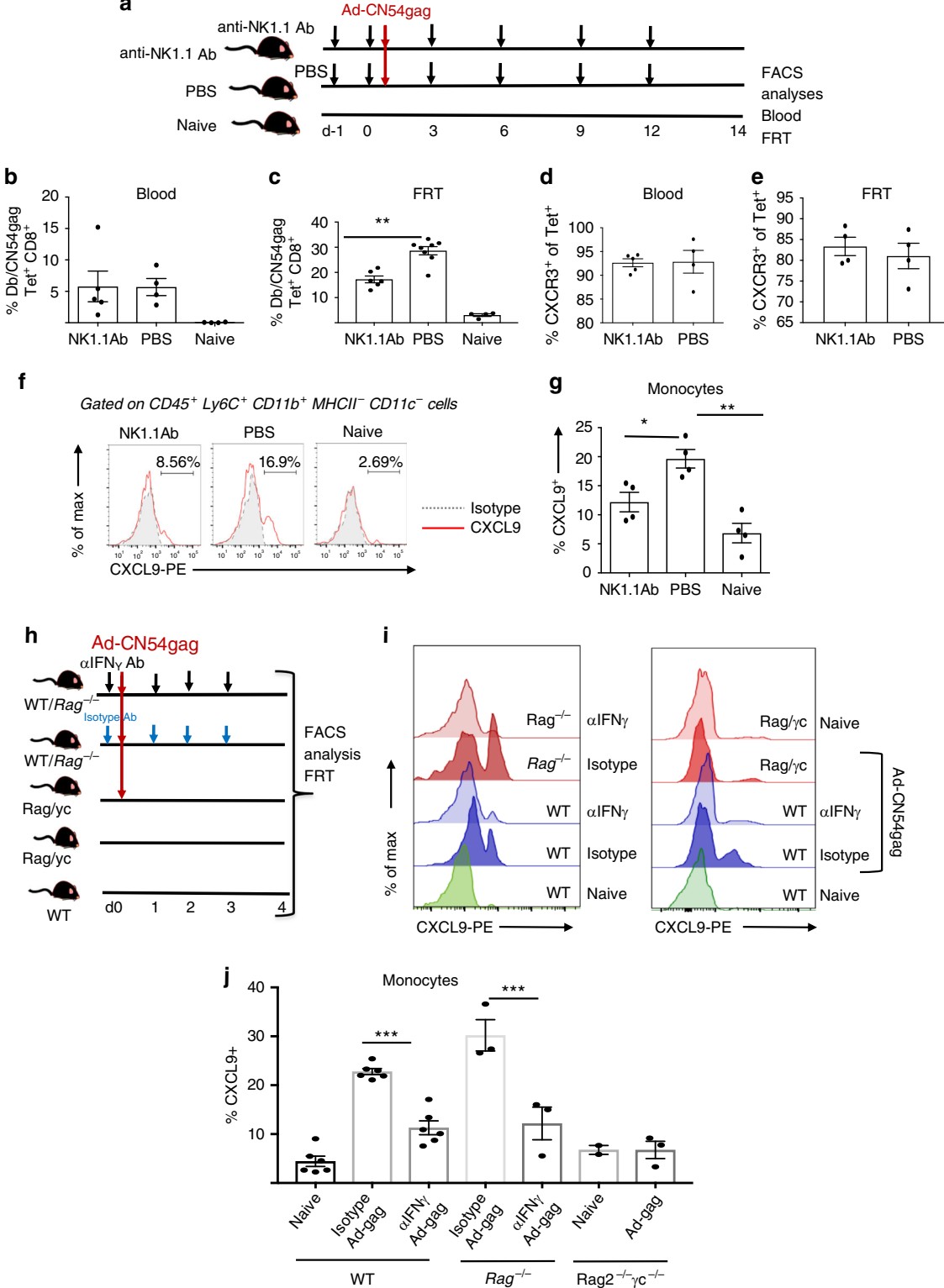

primed CXCR3-expressing CD8+ T cells to that tissue. Our study revealed that an innate lymphoid cell population in the local tissue could orchestrate CXCR3-dependent recruitment by responding rapidly and prime CXCL9 production by Ly6C^hi monocytes via cell intrinsic IFNγ. Together, our data suggest a general mechanism whereby ILC1 can regulate antigen-primed CXCR3+ CD8+ T cells to barrier tissues that are not the target site of immunisation. Importantly, we demonstrate that this

mechanism provides protection from VV pathogen challenge in the FRT.

It is currently understood that exposure of the vaginal mucosa to topically applied chemokines or to microbicides that create non-specific inflammation permits recruitment of systemically primed CD8+ T cells to the FRT[26,47]. This implies that signals derived from the microenvironment of epithelial barrier tissues can 'pull' circulating effector CD8+ T cells into the tissue and

**Fig. 8** Recruitment of CD8[+] T cells to the FRT requires group 1 ILCs and IFNγ. **a** The experimental design. Anti-NK1.1 antibody or PBS was injected i.p. 1 day before and at 3, 6, 9 and 12 days after skin immunisation by MA with Ad-CN54 gag. **b, c** Frequency of D[b]/CN54 gag Tet[+] CD8[+] T cells in peripheral blood (**b**) and FRT tissue (**c**) 14 days after immunisation. **d, e** Frequency of Tet[+] CD8[+] T cells expressing CXCR3[+] in peripheral blood (**d**) and FRT tissue (**e**) 14 days after immunisation. Data (**b–e**) are from two independent experiments ($n = 4$–8 per group). $**p < 0.01$ by one-way ANOVA, error bars show s.e.m. **f** Flow-cytometric analysis of CD45[+]Ly6C[+]CD11b[+]MHCII[−]CD11c[−] cells from each experimental group at 3 days post immunisation for CXCL9 (red histograms) and isotype control antibody staining (grey-filled histograms). **g** Summary graph showing CXCL9 expression from a gated CD45[+]Ly6C[+] CD11b[+]MHCII[−]CD11c[−] monocyte population. Data are from two independent experiments ($n = 4$–8 per group). $*p < 0.05$, $**p < 0.01$ by one-way ANOVA, error bars show s.e.m. **h** Experimental design for anti-IFNγ administration. WT or Rag1[−/−] (Rag[−/−]) were injected i.p. with anti-IFNγ antibody (500 μg) 1 day before and at day 0, 1, 2 and 3 relative to skin immunisation with Ad-CN54 gag. Rag2[−/−]γc-deficient (Rag[−/−]γc[−/−]) were immunised with Ad-CN54 gag or left naive. **i** Representative flow-cytometry profiles showing CXCL9 expression on monocytes (gated on CD45[+]Ly6C[+] CD11b[+]MHCII[−]CD11c[−]) from the FRT of each treatment group at day 4 post immunisation. Data are of $n = 3$ per group. $*p < 0.05$, $**p < 0.01$ by one-way ANOVA, error bars show s.e.m. **j** Summary bar graph showing frequency of CXCL9[+] Ly6C[hi] monocytes in FRT tissue isolated from each experimental group. Data are from two independent experiments, ($n = 2$–6 per group). $***p < 0.0001$ by one-way ANOVA, error bars show s.e.m

establish tissue residence[26]. Thus, neither direct immunisation nor on-going infection of the FRT are strictly essential for the establishment of memory CD8[+] T cells bearing hallmarks of CD8[+] T$_{RM}$ cells in the tissue[17,25,26,47]. Our data advance this concept by demonstrating that immunisation through the skin (and also by the IM route) with an Ad5 vaccine vector instructs peripheral tissues including the FRT and lung to recruit systemically generated CD8[+] T-cell effectors. In agreement with Nakanishi et al.[48], CD8[+] T-cell-expressed CXCR3 was absolutely essential for CD8[+] T-cell recruitment to the FRT. Nonetheless, the mechanism conditioning the FRT (and lung) to express the requisite cognate chemokine ligand differ, likely dictated by the pathogen/vaccine and site of pathogen/vaccine exposure. These different contexts might explain a seeming paradox that CD4[+] T-cell-induced IFNγ is essential for CXCL9-driven CD8[+] T-effector extravasation to the FRT and lung in respect of local HSV-2 and influenza virus infection, respectively[48,49]. In contrast, our study showed that NK1.1[+] ILC1 intrinsic IFNγ, but not CD4[+] T-cell-induced IFNγ, is necessary and sufficient to drive Ly6C[hi] monocyte-expressed CXCL9. This is essential for NK1.1-dependent CD8[+] effector T recruitment to the FRT where the initial Ad5 virus vector exposure is in the skin. Thus NK1.1[+] ILC1 can provide an IFNγ signalling module and licence CXCL9 production that resembles an effector function provided by CD4[+] Th1 cells[36,37,50], but differs from CD4[+] Th1 cells by the absence of TCR specificity in initiation of CXCL9-driven CD8[+] T effector extravasation. The licensing properties of ILC1 reported herein are consistent with the ability of activated NK cells to cluster with CD11b[+] cells[51,52] and Ly6C[hi] monocytes[44] during infection and prime their activation via IFNγ[44,53]. Therefore, Ly6C[hi] monocytes and CD11b[+] DCs are likely complementary as producers of a CXCL9 microenvironment necessary for recruitment of systemically primed CD8[+] effector T cells to epithelial barrier tissues. Our findings bare direct relevance to optimising delivery of vaccines where local immune surveillance by recruited CD8[+] T cells plays a critical role in protection at the epithelial barrier[3,54]. In contrast, vaccines that provide antibody-based protection at the cervico-vaginal epithelia, exemplified by the IM-administered prophylactic HPV sub-unit vaccine, have been proposed to work by a different mechanism, transudation or exudation of serum IgG as opposed to providing signals to recruit antibody-secreting cells to the epithelia[55].

An outstanding question remains, how do ILC1 within distant epithelial barrier tissues proliferate and undergo activation that conditions the FTR to recruit CD8[+] T effectors in response to skin immunisation? Immune connectivity across distant epithelial barrier tissues has been revealed previously following skin vaccination[17,25,26,47,56]. This may reflect the action of type I IFN[56] or other innate mediators responding at the site of immunisation[57], acting systemically that may be essential for

ILC1 activation. Unlike MVA[56] or adjuvanted vaccines[57], rAd5, used in this study, is a weak inducer of serum IFNα[28,58]. Equally, type I IFN-dependent activation of NK cells is not a property of rAd5[58], thus other mechanisms must operate. Intriguingly, our study found a low but consistent copy number of vaccine vector genomes in peripheral non-lymphoid tissues (including the FRT) after skin immunisation, consistent with previous reports[59–61]. The biodistributed Ad5 vector dose in this study had the capacity to recruit populations of ILC1, Ly6C[hi] monocytes and CXCR3[hi] CD8[+] T effector cells to the FRT after intravaginal injection of naive mice. This leads us to speculate that low levels of virus vector may bypass subcapsular sinus macrophages in skin-draining LN after immunisation and disseminate systemically[62]. There is a precedence that blood monocytes can rapidly interact with Ad5[63] and are highly suited to hijack and transport virus or virus genome via the blood circulation to distal tissues[64,65]. Equally, monocytes are acknowledged to enter extravascular tissues and replenish resident monocyte-derived cells under steady-state conditions[66,67]. This might be one mechanism by which crosstalk between monocytes carrying virus/vector genome and tissue-resident ILC initiate a feedback loop to recruit waves of monocytes and ILC1 that may condition the FTR in the context of skin immunisation. Although our study with Ad5-vectored vaccines emphasised local activation of innate NK 1.1[+] cells and their role in orchestrating recruitment of CD8[+] T effector and CD8[+] T$_{RM}$ cells to the FRT after skin immunisation, our findings do not preclude if this response state of NK1.1[+] cells might also be dictated in part by cytokines acting systemically from the site of immunisation[68]. Additional studies with other vaccine vectors and forms of immunogens will be important.

In conclusion, we have defined a mechanism whereby skin vaccination induces low-level inflammation in the FRT that is sufficient to promote CD8[+] T-cell recruitment through a CXCL9-dependent mechanism licensed by ILC1 cells. These data may have important implications for vaccine designs against pathogens transmitted across epithelial barrier tissues and highlight the attributes of the skin as a delivery site to stimulate prophylactic and therapeutic immunity against vaginally acquired infections.

## Methods

**Recombinant adenovirus vaccines.** Replication-deficient, E1, E3-deleted adenovirus type 5 (Ad5) vaccine vectors were propagated in low-passage human embryo kidney 293 (293AD) cells (Cambridge Biosciences) at a multiplicity of infection of 25 virus particles per cell (as titrated by the DNA Pico-Green assay, Invitrogen) and then purified by the Native Antigen Company. Viruses were stored at −80 °C until use. rAd5-HIV-1 CN54 gag encodes a codon optimised synthetic HIV-1 CN54 gag gene reported previously[27]. Ad5-OVA encodes a non-secreted chicken ovalbumin gene, constructed by Dr M. Zenke's laboratory (Aachen University, Aachen, Germany)[27]. Virus particle titres were determined by the DNA Pico-Green assay (Invitrogen).

**Recombinant vaccinia virus HIV-1 CN54 gag**. A codon optimised synthetic HIV-1 CN54 gag gene sequence (Supplementary Fig. 14) was amplified using HIV-1 CN54 gag primers (Supplementary Table 1) using high-fidelity PCR by Invitrogen pFx polymerase and inserted into a plasmid containing vaccinia virus thymidine kinase flanking arms (pVACV-TK). The recombinant plasmid was sequenced to confirm insertion and sequence fidelity. Sub-confluent BHK cells ($2 \times 10^6$ cells) were transfected with 2 μg of pVACV-TK-CN54gag using Lipofectamine and Plus Reagent (Invitrogen) and co-infected with parental VACV:GyrB-PKR at an MOI of 0.05 pfu/cell. At 30 h post infection, cells were scraped and virus released by three rounds of freeze–thaw treatment followed by sonication. RK13 cells pre-treated with coumermycin A1 (100 ng/ml) for 16 h were infected with dilutions of the in vitro recombined virus extract and overlaid with media containing coumermycin. Thirty hours post infection, plaques were isolated and amplified under selection with cumeromycin in RK13 cells. Amplified virus was purified by centrifugation through 36% sucrose. Virus was maintained under selection at all times, and sequences were confirmed by Arizona State University core sequencing laboratory.

**Fabrication of microneedle arrays (MA)**. Dissolvable microneedles (1500 μm in length, 670 μm in base diameter and 44 per array) were fabricated by a centrifugation casting method (at 1780×g for 1 min) using an inverted cone-shaped silicone template. Vaccine vectors were formulated in the matrix of the needle tips at a 1:1 ratio with sodium carboxyl methylcellulose (8% wt/vol Na-CMC) and sucrose (30% wt/vol). A second layered matrix (12% Na-CMC, 4.8% lactose) created the needle shaft and a pre-made membrane (8% Na-CMC, 0.8% lactose) formed the needle base. After air drying (24 h at room temperature), the MAs were carefully removed from the template and stored in a desiccator at room temperature.

**Mice**. Female mice at 7–8 weeks of age were used in this study. C57BL/6 mice were purchased from Envigo. Rag$^{-/-}$ OT-I mice on a CD45.1 background (B6.SJL CD45.1) were from The Francis Crick Institute (London) and Rag1$^{-/-}$ and Rag2$^{-/-}$γc$^{null}$ mice were bred at King's College London. The minimum numbers of mice required to obtain statistically significant and reliable results were used. The number of animals within each study arm is denoted within the appropriate figure legends.

**Ethics statement**. All animal husbandry and experimentation were approved by King's College London ethics committee and performed under a project license granted by the United Kingdom Home Office.

**Depo-Provera synchronisation**. All mice in this study received medroxyprogesterone acetate, Depo-Provera (Depo®, Pfizer) at a dose of 3 mg by s.c. injection 5 days before each experiment.

**Immunisation models**. Mice received either $1 \times 10^9$ vp (or where indicated $1 \times 10^7$ vp) of rAd5 vaccine vector either by MA administration, where MAs were applied manually with gentle pressure (5 min) to the shaved dorsal surface of the ear or back skin (as indicated) or by ID or IM injection. In some experiments, mice received the designated dose (or a lower dose, where indicated) of rAd5 vaccine vector by injection directly into the vaginal wall.

**Adoptive transfer**. Naive donor antigen-specific CD8$^+$ T cells were isolated from the spleens of CD45.1$^+$ transgenic OT-I mice and magnetically purified (>96%) using a CD8 T cell isolation kit (Stemcell Technologies). For effector cell generation, $2 \times 10^5$ naive CD45.1 OT-I CD8$^+$ donor T cells were adoptively transferred into recipient B6 mice. The next day, recipients were immunised ID with $1 \times 10^9$ vp of rAd5-OVA. Some recipient mice were injected i.p. with a blocking antibody against CXCR3 (200 μg, clone: CXCR3–173, 2BScientific) at day 6 post immunisation. FACS analysis confirmed CXCR3 depletion (>99.5%). Effector OT-I cells (either CXCR3 depleted or not) were isolated from the spleen at 7 days post immunisation. Single-cell suspensions were purified using the MagniSort$^{TM}$ mouse CD45.1 positive selection kit, and then $2 \times 10^6$ cells were transferred i.v into naive hosts or into secondary recipients immunised 3.5 days previous to cell transfer with either rAd5-OVA (by skin MA or by intravaginal immunisation) or with rAd5-HIV-1 CN54 gag or with PBS by intravaginal immunisation (as indicated). On the day of cell transfer, recipients of CXCR3 blocked CD45.1 OT-I also received an i.p. injection of 200 μg of anti-CXCR3 antibody (clone: CXCR3–173, 2BScientific). After one and a half days, the numbers of CD45.1 OT-I cells harvested from the blood, spleen and FRT of naive and secondary recipients were analysed by flow cytometry.

**Isolation of cells from tissues**. At various time points, single-cell suspensions were prepared from blood, spleen and LNs and re-suspended in complete RPMI [RPMI medium 1640 supplemented with 100 U/mL penicillin, 100 μg/mL streptomycin, 2 mM L-glutamine (Invitrogen) and 10% heat-inactivated FBS (Biosera). The liver, lung and FRT (where indicated) were harvested following perfusion. Single-cell suspensions from the FRT and lung were obtained by enzymatic digestion for 30 min at 37 °C using 1 mg/mL Collagenase D (Roche$^{TM}$) and 0.02 mg/mL DNase I from bovine pancreas (Roche$^{TM}$).

**Flow cytometry**. The following antibodies (including clone and catalogue number) were purchased from BD Bio-sciences: purified anti-FcRII/III mAb (CD16/32, 2.4G2), anti-CD8 (53–6.7), anti-CD4 (RM4–5), anti-CD11b (M1/70), anti-CD3 (145–2C11), anti-MHC class II (2G9), anti-Gr-1 (RB6–8C5), anti-CD45R/B220 (RA3–6B2), anti-CD25 (7D4), anti-CD4 (GK1.5), anti-CD115 (AFS98), anti-NKp46 (29A1.4), anti-CD49b (DX5), anti-CCR1 (S15040E), anti-CCR5 (HM-CCR5), anti-CCR6 (29–2L17), anti-CCR7 (4B12), anti-CCR9 (9B1), anti-CCR10 (6588–5), anti-CXCR3 (CXCR3–173), anti-CXCR6 (SA051D1), anti-CXCL9 (MIG-2F5.5), anti-CD11c (HL3) and anti-α4β7 (DATK32). In addition, anti-IFNγ (XMG1.2), anti-Granzyme B (NGZB), anti-Ly6C (HK1.4), anti-CD69 (H1.2F3) and anti-CD11a (M17/4) were obtained from eBioscience. Anti-NKp46 (29A1.4), anti-NK1.1 (PK136), anti-CD45 (30F-11), anti-CD8 (53–6.7), anti-CD45.1 (A20), anti-CD45.2 (104), anti-CD11c (N418) and anti-MHCII (M5/114.15.2) were acquired from Biolegend. Dilutions at which antibodies were used are shown in Supplementary Table 2. Tetramers to the immunodominant H-2D$^b$-restricted HIV-1 CN54 gag 308–318 epitope (D$^b$/CN54gag) were kindly synthesised by the NIH Tetramer core facility (Emory University, USA). Cells were stained with D$^b$/CN54gag tetramer (15 min) at room temperature, followed by addition of cell-surface antibodies (20 min) and then re-washed. For intracellular cytokine staining, $1 \times 10^6$ cells per mL were incubated 6 h at 37 °C with anti-CD28 (2 μg/mL) either alone (unstimulated control) or with peptide HIV-1 CN54 gag $_{308–318}$ (2 μg/mL). Brefeldin A (10 μg/mL) (Sigma-Aldrich) was added for the last 5 h of culture. After washing, cells were incubated with anti-FcγRII/III (10 min), followed by staining with D$^b$/CN54gag tetramer (15 min) prior to incubation with antibodies to cell-surface markers (20 min). Cells were fixed and permeabilized with the BD Cytofix/Cytoperm Kit according to the manufacturers' instructions and then stained simultaneously with the D$^b$/CN54gag tetramer, anti-IFNγ and anti-Granzyme B antibodies (20 min). Samples were acquired using a BD FACSCanto II or BD SORP LSRFortessa$^{TM}$ flow cytometer and analysed using FlowJo software version 9.7.5 (Tree Star) for Mac. Gating strategies to track D$^b$/CN54gag Tet$^+$ CD8$^+$ T cells in blood, LN, FRT, lung and spleen are shown in Supplementary Fig. 15. Gating strategy to define (i) Ly6C$^{hi}$ monocytes, DCs and NK1.1+ (ILC1) cells in blood and FRT; (ii) to examine CXCL9 expression by Ly6C$^{hi}$ monocytes and DCs in blood, FRT and lung and (iii) to examine CD45.1$^+$ OT-1 cells following adoptive transfer in tissues are shown in Supplementary Fig 16.

**Quantitative real-time PCR for adenovirus copy number**. Quantitation of Ad5 copy number in tissues after MA immunisation was conducted by using a real-time TaqMan RT-PCR system. DNA was purified from collected tissues using a Gene EluteTM Mammalian Genomic DNA kit (SIGMA-ALDRICH), according to the manufacturer's instructions. The purified DNA was mixed with TaqMan PreAmp Master Mix (2x) (Applied Biosystem), the primer pair (Supplementary Table 1) and TaqMan Probe FAM-taccagaacgaccacagca-MGB (total volume per reaction: 10 μl). The mixture was amplified in an automated fluorometer, QuantStudio5 (Applied Biosystem) under the following conditions: 40 cycles of 95 °C for 15 s and 60 °C for 60 s. rAd5-HIV-1 CN54 gag was used as a standard.

**Virus challenge and plaque assay**. Groups of mice were challenged by depositing $5 \times 10^7$ vp of rVV-CN54gag into the vaginal cavity and monitored for 6 days. FRT tissues were harvested at 6 days post rVV-CN54gag challenge and homogenised by enzymatic digestion for 30 min at 37 °C using 1 mg/ml Collagenase D (Roche$^{TM}$). The supernatants were serially diluted ($10^{-2}$ to $10^{-7}$) in the Dulbecco's modified Eagle's medium (DMEM) and added onto BSC-40 cell monolayers for 1 h at 37 °C. Cells were then cultured in complete DMEM containing 5% heat-inactivated FBS and containing coumermycin A1 (100 ng/ml) for 48 h before fixation an de-staining with 0.1% crystal violet solution for counting viral pfu.

**In vivo killing assay**. Naive syngeneic splenocytes were labelled with CFSE at 5 μM (CFSE$^{hi}$) or 0.5 μM (CFSE$^{lo}$), pulsed with or without peptide to the immunodominant Db restricted HIV-1 CN54 gag epitope (5 μg/ml), then $1 \times 10^6$ cells (mixed at a 1:1 ratio) injected using a Hamilton syringe with 30G needle into the vaginal wall of immunised or naive control mice and harvested after 24 h to measure in vivo cytolysis of target cells by the loss of the CFSE$^{hi}$ peptide-pulsed population relative to the control CFSE$^{lo}$ population by flow cytometry.

**In vivo antibody-dependent cell depletion or neutralisation**. For depletion of NK1.1 expressing innate cells, mice were injected with 200 μg of purified monoclonal NK1.1 antibody (PK136, BioXcell) i.p. 24 h before immunisation, followed by injections on day 0, 3, 6, 9 and 12. For IFNγ neutralisation, mice were injected i.p. with purified anti-IFNγ antibody (XMG1.2, BioXcell) 24 h before and daily after immunisation. Control mice received their corresponding isotype Ab.

**Statistical analysis**. The results were analysed with GraphPad PRISM$^{TM}$ version 6.0 (San Diego, CA, USA). Bars in figures show the mean ± SEM. Comparisons between groups were performed using a one-way ANOVA in conjunction with

Bonferroni post analyses test. Probability values are expressed as the following: ***$p < 0.001$, **$p < 0.01$ and *$p < 0.05$.

**Reporting summary**. Further information on research design is available in the Nature Research Reporting Summary linked to this article.

## Data availability

All relevant data are available from the authors upon reasonable request. A reporting summary for this article is available as a Supplementary Information file. The source data underlying all reported averages in graphs underlying Figs. 1c, d, g, 2c, g, e, 3a–d, 4a, c–e, 5a–c, 6a–b, 7b–e, 8b–e and j and Supplementary Figs. 1, 2, 6, 8, 9 and 11 are provided as a Source Data file.

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

## Acknowledgements
We thank the NIH Tetramer facility (Emory University) for kindly providing the tetramers used in this study, Professor Ralf Wagner, University of Regensburg for providing a codon optimised synthetic HIV-1 CN54 gag gene, Dr Ian Rosswell, Francis Crick Institute for OT1 B6.SJL CD45.1 mice and the support staff at the Flow Cytometry Core Facility, Programme in Infection and Immunity, King's College London. This work was funded by The Bill and Melinda Gates Foundation, Seattle, WA grant number 38639 and European Union Marie Curie Initial Training Network (UniVacFlu) grant number 607690 to LSK and grants BB/L027933/2 and CH/11/2/28733 to AHB.

## Author contributions
M.Z. and L.S.K. designed the research; M.Z., P.D.B., C.H., P.K., A.D., B.I.Y., L.A.O.'N. and C.C. performed the research; N.B. and S.-Y.K. contributed reagents/materials; M.Z., P.D.B., C.H., A.D., P.K., B.I.Y., L.A.O.'N., G.M.L. and L.S.K. analysed the data, M.Z. and L.A.O.'N. generated the figures, L.S.K. wrote the paper. All authors contributed to editing the paper.

## Additional information

**Competing interests:** The authors declare no competing interests.

