## [Peer Review File · Nature Communications]

Reviewers' comments:

Reviewer #1 (Remarks to the Author):

M. Zaric et al. investigated the mechanism of CD8⁺ effector T cell recruitment into mucosal tissue. They report that adenovirus serotype 5 (Ad 5) bio-distributes at very low level to non-lymphoid tissues (including female reproductive tract, FRT) after skin immunization. Virus distribution to FRT triggers activation and expansion of NK1.1⁺ group 1 innate lymphoid cells (ILC1). IFN γ produced by ILC1 triggers the production of the chemokine CXCL9 by CD11b⁺Ly6C⁺ monocytes. CXCL9 is required to recruit skin primed CXCR3⁺ CD8⁺ T-cells to the FRT.

Results are from a combination of various in vivo and in vitro experimental approaches. The study is interesting and addresses an important unresolved question in immunology concerning the mechanism by which immunization in a localized skin area may result in CD8⁺ T cell mediated intra-mucosal response in different districts of the organism. However, the study has, to my opinion, some weak points.

1) One general concern I have is about the pathophysiological relevance of the reported results. The observations that virus bio-distributes at very low level to non-lymphoid tissues (in principle everywhere in the body) and that CD8⁺ T cell recruitment to the mucosa is independent of their antigen specificity (Fig. 4e) are puzzling. They suggest that, during viral infection CXCR3⁺ activated T cells (either antigen specific or not) might redistribute to various districts even where they are not useful. One explanation of the results shown in Figure 4e is that transferred CD45.1 cells might recognize antigens expressed by the viral vector (independently of OVA). This possibility should be definitively ruled out.

2) Results shown in Figure 2 are not fully convincing. While panel 2b shows that specific target cells are decreased in number when compared to control target cells, it does not directly prove that these cells have been killed by Db/CN54gag Tetramer⁺ CD8⁺ T cells. The ex vivo functional characterization of these cells shown in Figure 2F is not fully convincing. The percentage of GrzB⁺ T cells is similar in naive, MA immunized and ID immunized mice. The percentage of IFN γ ⁺ T cells is moderately increased in previously immunized mice. To address this point it is important to investigate whether or not isolated Db/CN54gag Tetramer⁺ CD8⁺ T cells can kill cognate target cells in vitro.

3) The experiments shown in Figure 4 (based on the transfer of congenic CD45.1 T cells) are interesting and allow to investigate the impact that blocking of CXCR3 has on antigen specific CD8⁺ T cell recruitment. However the employed approach is somehow indirect. What is the impact of anti-CXCR3 blocking antibodies on CD8⁺ T cell recruitment to mucosa in Ad-CN54gag skin (MA or ID) immunized mice?

In Figure 4 one control is missing if I am not wrong. To support the conclusion that CXCR3 is the key chemokine receptor involved in activated CD8⁺ T cell recruitment, the authors should show the level of CXCR3 expression by CD8⁺ Tetramer negative T cells both in naive and skin immunized mice.

4) Figure 6 shows that the recruitment of CD11b⁺Ly6C⁺ monocytes into FRT is delayed and sustained up to 7 days post skin immunization. To support the model proposed by the authors it is important to define whether CD11b⁺Ly6C⁺ monocytes are present in FRT when the CD8⁺ recruitment is measured (14 days). This point is also valid for results shown in Figure 5: CXCL9 expression should be investigated at 14 days.

Reviewer #2 (Remarks to the Author):

The MS by Zaric et al is an likely important report on the role of skin immunization with an Ad vector to drive immunity ultimately at the female reproductive tract. The studies are interesting and are mostly supported by detailed examination of immune cell activation which results in identifying a novel mechanism by which skin immunization appears in this case to generate immune cell trafficking to the FRT. The trafficking is associated with improved impact against mucosal viral challenge in a model challenge system. Two different skin immunization strategies were studied that performed mostly equally well. These studies provide an argument for a novel role of CD3- NK1.1 - group1 innate lymphoid cells (ILC1 cells) having an important role in skin immunization, which has importance to this field in general. The studies are also novel in that the authors report protection at epithelial barriers by the first responder cells is not only due to interferon gamma having direct activity on limiting viral replication and dissemination, but also that ILC1 derived interferon gamma is able to engender inflammatory monocytes to secrete CXCL9, which helps draw CD8 T cells to the cite which are then capable of impacting viral challenge.

In this paper, the authors show that irrespective of precise location or modality, skin immunization with a replication defective Ad5 virus expressing a HIV Gag antigen is able biodistribute to the FRT at low numbers. This in turn draws antigen specific CD8 effector T cells to the female reproductive tract through expansion and activation of ILC1 cells. These ILC1 cells secrete interferon gamma which then licenses Ly6C+ monocytes to produce CXCL9, a chemokine that recruits CXCR3+ CD8 T cells to the female reproductive tract. These CD8 T cells were able to protect against a viral vaginal challenge and resulted in an increased amount of killing of cells expressing Gag. Depletion of the ILC1 cells was associated with a decrease in the number of skin primed CXCR3+ CD8 T cells, underscoring their necessity in this alternative mechanism for protection in the FRT.

1. An important control is to examine intramuscular delivery as well as formal proof that that there would be differences that are not due uniquely to micro needles for example seeding with the FRT with CXCR3+ CD8 T cells should be examined.

2. Viral titres were impacted when skin immunized mice were challenged 4 weeks post immunization. Do the authors know how durable this response is? Do they have evidence that it is longer lived than 4 weeks? Do the CD8T cells that are attracted progress to becoming resident memory T cells in the FRT to continue their surveillance against future infection.

3. Given that the ability of the Ad5 virus to seed the FRT when administered intravaginally was dose dependent and that $3.5E2$ vp could not elicit these CD8 T cells, whereas $1E9$ vp did generate CD8 in FRT, this suggests that there could be a dose dependence in the skin as well? Do the authors have evidence for such a dose response?

4. The authors showed the connection of induction of CXCL9 producing inflammatory Ly6C monocytes to aid in the recruitment of CXCR3+ CD8 T cells, but they did not address the contribution of the dendritic cells which can also express increased CXCL9. Do the DC themselves have a role in the recruitment?

5- this data seems to particularly be important in protection by CD8 T cells, as the HPV prophylactic vaccines appear to protect from infection at the mucosa with high efficacy and these

are applied by the IM route. These vaccines are however associated with Ab based protection. Perhaps the authors should consider mentioning this difference in the discussion section.

There a few grammatical errors that should be addressed in the intro section and in the discussion.

Manuscript NCOMMS-18-28357: Response to Reviewer's Comments

We would like to thank both reviewers for their time, comments and suggestions, which we believe, have significantly strengthened our manuscript. Changes to the revised manuscript text are indicated with yellow highlighting for ease of re-review.

Reviewer #1 (Remarks to author):

M. Zaric et al. investigated the mechanism of CD8⁺ effector T cell recruitment into mucosal tissue. They report that adenovirus serotype 5 (Ad 5) bio-distributes at very low level to non-lymphoid tissues (including female reproductive tract, FRT) after skin immunization. Virus distribution to FRT triggers activation and expansion of NK1.1⁺ group 1 innate lymphoid cells (ILC1). IFN γ produced by ILC1 triggers the production of the chemokine CXCL9 by CD11b⁺Ly6C⁺ monocytes. CXCL9 is required to recruit skin primed CXCR3⁺ CD8⁺ T-cells to the FRT.

Results are from a combination of various in vivo and in vitro experimental approaches. The study is interesting and addresses an important unresolved question in immunology concerning the mechanism by which immunization in a localized skin area may result in CD8⁺ T cell mediated intra-mucosal response in different districts of the organism.

However, the study has, to my opinion, some weak points.

Specific comments

1) One general concern I have is about the pathophysiological relevance of the reported results. The observations that virus bio-distributes at very low level to non-lymphoid tissues (in principle everywhere in the body) and that CD8⁺ T cell recruitment to the mucosa is independent of their antigen specificity (Fig. 4e) are puzzling. They suggest that, during viral infection CXCR3⁺ activated T cells (either antigen specific or not) might redistribute to various districts even where they are not useful. One explanation of the results shown in Figure 4e is that transferred CD45.1 cells might recognize antigens expressed by the viral vector (independently of OVA). This possibility should be definitively ruled out.

Author response:

In figure 4e, we quantitated the number of adoptively transferred CD45.1 Rag^{-/-} OT1 TCR Transgenic cells that were recruited to the FRT of congenic CD45.2 recipients. The TCR of the transferred Rag-deficient CD45.1 cells in this experimental model is specific to the OVA 257-264 epitope. By enumeration on CD45.1 using a monoclonal antibody specific to mouse CD45.1, the data in Figure 4e represents the absolute number of OVA 257-264 specific TCR transgenic OT1 T cells recruited to the FRT of CD45.2 recipients. Therefore the possibility that the transferred CD45.1 cells might recognise antigens expressed by the viral vector independently of OVA is ruled out by the experimental design.

We have revised the legend for Figure 4 to make it clear that Rag deficient CD45.1 OT-1 effector cells were transferred and enumerated for CD45.1⁺ expression.

2) Results shown in Figure 2 are not fully convincing. While panel 2b shows that specific target cells are decreased in number when compared to control target cells, it does not directly prove that these cells have been killed by Db/CN54gag Tetramer+ CD8+ T cells. The ex vivo functional characterization of these cells shown in Figure 2F is not fully convincing. The percentage of GrzB+ T cells is similar in naive, MA immunized and ID immunized mice. The percentage of IFN γ + T cells is moderately increased in previously immunized mice. To address this point it is important to investigate whether or not isolated Db/CN54gag Tetramer+ CD8+ T cells can kill cognate target cells in vitro.

Author response:

We chose to quantitate the killing capacity of skin primed CD8+ T effectors in the FRT using the established protocol in the field; relative loss of peptide specific target cells in comparison with the control targets (Barber D, Whery J and Ahmed R, J.I 2003) to definitively indicate peptide-specific killing in vivo. We would like to highlight that the total number of CD8+ T cells in the FTR (unlike from spleen or lymph nodes) is too low for isolated cells to be used in the *in vitro* killing assay suggested by the reviewer. The *in vitro* assay requires at a minimum, 1×10^6 effector cells where e.g. 2.5×10^4 target cells are incubated at an effector to target cell ratio of 2.5:1, 5:1 and 10:1 in duplicate. However, the mean total number of Db/CN54gag Tetramer+ CD8+ T-cells detected in the FRT of skin-immunised mice from a representative experiment, where n=12, is $1.7 \times 10^3 \pm 0.19 \times 10^3$ (Figure R1). This value is comparable with data published previously (in other class I tetramer systems in the FRT) by the labs of Dr Iwasaki (Nakanishi Y et al. Nature, 2009), Dr Berzofsky (Wang Y et al. Nat Comms. 2015) and Dr Schiller (Cuburu N et al. Int. J Cancer 2018). Thus, due to cell number, it is technically not possible to investigate whether or not isolated Db/CN54gag Tetramer+ CD8+ T cells from the FRT can kill cognate target cells *in vitro*.

For clarity, we have indicated in the legend (Figure 2) and in the results section that the HIV-1 CN54 gag peptide used to pulse syngeneic target cells in the *in vivo* killing assay and the peptide folded in the Tetramer (Db/CN54gag) used for tracking studies are both specific to the immunodominant Db restricted HIV-1 CN54 gag epitope.

Figure R1: Total number of Db/CN54gag Tetramer+ CD8+ T cells in the FRT at 14 days post skin immunisation. C57BL/6 mice were immunised with Ad-CN54 gag (1×10^9 vp) by the intradermal route. Data shows the absolute number of CD8+ Db/CN54 gag tetramer+ cells (n= 12 mice / experimental group), error bars show SEM.

As suggested by the reviewer, we have replaced the original Fig 2f with a new data set shown in Fig 2f (of the revised manuscript) that in our opinion distinctly demonstrates that at day 14 post skin immunisation Tetramer+ CD8+ T cells recruited to the FRT express intracellular interferon-gamma and GrzB following in vitro stimulation with peptide HIV-1 CN54 gag 308-319. Re-stimulated cells were analysed by gating on a CD45+MHCII-CD4-CD8+ population, then a tetramer gate (surface and intracellular stain) was examined for IFN γ + and GrzB+ cells.

3) The experiments shown in Figure 4 (based on the transfer of congenic CD45.1 T cells) are interesting and allow to investigate the impact that blocking of CXCR3 has on antigen specific CD8+ T cell recruitment. However the employed approach is somehow indirect. What is the impact of anti-CXCR3 blocking antibodies on CD8+ T cell recruitment to mucosa in Ad-CN54gag skin (MA or ID) immunized mice?

To provide compelling evidence for the role CD8+ T cell expressed CXCR3 provides in recruitment of systemically primed CD8+ T cells to the FRT, we specifically opted to use an adoptive transfer model in which donor CD8+ T-cell effectors were transferred into congenic skin-immunised recipients that received anti-CXCR3 blocking antibody. This approach was technically more challenging and in our opinion cleaner than treating skin-immunised mice directly throughout the experiment with anti-CXCR3 blocking antibody. Our experimental design specifically enabled us to exclude the effect CXCR3 blocking antibodies may have on the migration and positioning of many cell types within lymph node compartments that are essential to provide signals required for the generation and expansion of effectors from naïve CD8+ T cells. CXCR3 is expressed early during the differentiation of CD8+ effector precursors (Kurachi M et al JEM 2011, Shah S et al 2015 eLife). Blocking by antibody treatment or by genetic deficiency is reported to prevent CXCR3 enabled CD8+ T cells to cluster with IL-12 producing DCs to complete their differentiation (Shah S et al eLife 2015) and to prevent acquisition of signals essential for their proliferation (Hu JK et al PNAS 2011). Likewise, CD4+ T cells that are reported essential to provide T-cell help for Ad vector primed CD8 T-cells (Provine NM et al JI 2016) require CXCR3 ligand cues to migrate to DCs for their activation (Groom JR et al. Immunity, 2012).

Therefore, for all the aforementioned reasons to exclude the effects of CXCR3 blocking antibodies in the priming of CD8+ T cell effectors from the role CXCR3 performs in migration of CD8+ T cell effectors to the FRT, our approach was to generate fully functional CD8+ effector T cells in donor mice and then transfer to recipients injected with anti-CXCR3 blocking antibody. For these considerations we believe our approach provided a definitive answer to the question we addressed,

and why injecting anti-CXCR3 blocking antibodies throughout the course of the response to immunisation would not.

In Figure 4 one control is missing if I am not wrong. To support the conclusion that CXCR3 is the key chemokine receptor involved in activated CD8+ T cell recruitment, the authors should show the level of CXCR3 expression by CD8+ Tetramer negative T cells both in naive and skin immunized mice.

As suggested by the reviewer, we have further analysed the data to quantitate CXCR3 expression by tetramer negative CD8+ T cells in naive and skin immunized mice (see new Supplementary Figure 6). We find a small number of tetramer negative CD8+ T cells express CXCR3 (348 ± 75) in the FRT of naïve mice. While, in skin immunised there is an increased number of CXCR3+ CD8+ tetramer negative cells (1141 ± 134) compared with naïve mice. This is not unexpected. Figure 1f demonstrates a population of activated CD8+ T cells within the Db/CN54 gag tetramer negative population of skin-immunised mice as revealed by CD11a+ expression. These CD11a+ Db/CN54 gag 308-318 tetramer negative CD8+ T cells recruited to the FRT likely contain TCR specificities to other gag epitopes primed by Ad CN54 gag, that we have reported previously (Chowell D et al PNAS 2015) and also to the viral vector (e.g Db/FL9 Adenoviral epitope, our unpublished data). This point is discussed in the revised manuscript.

4) Figure 6 shows that the recruitment of CD11b+Ly6C+ monocytes into FRT is delayed and sustained up to 7 days post skin immunization. To support the model proposed by the authors it is important to define whether CD11b+Ly6C+ monocytes are present in FRT when the CD8+ recruitment is measured (14 days). This point is also valid for results shown in Figure 5: CXCL9 expression should be investigated at 14 days.

As suggested by the reviewer, we have now examined the presence of CD11b+Ly6C+ monocytes in the FRT and CXCL9 expression by this population from skin immunised and naïve mice (see new Supplementary Figure 9 and Figure 11). The frequency of CXCL9+ CD11b+ Ly6C+ monocytes at day 14 relative to the incidence in naïve mice ($p < 0.01$) supports the model of an innate-monocyte axis regulating CD8+ T cell recruitment to the FRT via CXCL9.

Reviewer #2 (Remarks to the Author):

The MS by Zaric et al is an likely important report on the role of skin immunization with an Ad vector to drive immunity ultimately at the female reproductive tract. The studies are interesting and are mostly supported by detailed examination of immune cell activation which results in identifying a novel mechanism by which skin immunization appears in this case to generate immune cell trafficking to the FRT. The trafficking is associated with improved impact against mucosal viral challenge in a model challenge system. Two different skin immunization strategies were studied that performed mostly equally well. These studies provide an argument for a novel

role of CD3- NK1.1 - group1 innate lymphoid cells (ILC1 cells) having an important role in skin immunization, which has importance to this field in general. The studies are also novel in that the authors report protection at epithelial barriers by the first responder cells is not only due to interferon gamma having direct activity on limiting viral replication and dissemination, but also that ILC1 derived interferon gamma is able to engender inflammatory monocytes to secrete CXCL9, which helps draw CD8 T cells to the site which are then capable of impacting viral challenge.

In this paper, the authors show that irrespective of precise location or modality, skin immunization with a replication defective Ad5 virus expressing a HIV Gag antigen is able to biodistribute to the FRT at low numbers. This in turn draws antigen specific CD8 effector T cells to the female reproductive tract through expansion and activation of ILC1 cells. These ILC1 cells secrete interferon gamma which then licenses Ly6C+ monocytes to produce CXCL9, a chemokine that recruits CXCR3+ CD8 T cells to the female reproductive tract. These CD8 T cells were able to protect against a viral vaginal challenge and resulted in an increased amount of killing of cells expressing Gag. Depletion of the ILC1 cells was associated with a decrease in the number of skin primed CXCR3+ CD8 T cells, underscoring their necessity in this alternative mechanism for protection in the FRT.

Specific comments

1. An important control is to examine intramuscular delivery as well as formal proof that there would be differences that are not due uniquely to micro needles for example seeding with the FRT with CXCR3+ CD8 T cells should be examined.

We have now examined recruitment of Db/CN54 gag tetramer+ CD8+ T cells to the FRT after intramuscular delivery and cross compared with recruitment after skin immunisation (see new Supplementary Figure 1). We find that recruitment of CD8+ T cell effectors to the FRT is not uniquely due to skin delivery of a viral vector by the intradermal or microneedle route. This may suggest that the innate lymphoid cell-monocyte axis we have identified in the context of skin delivery is also activated by other systemic routes and that this mechanism serves to establish CD8+ T cell surveillance at barrier tissues. We have revised the text to include this point.

2. Viral titres were impacted when skin immunized mice were challenged 4 weeks post immunization. Do the authors know how durable this response is? Do they have evidence that it is longer lived than 4 weeks? Do the CD8T cells that are attracted progress to becoming resident memory T cells in the FRT to continue their surveillance against future infection.

This is indeed an important point. We have previously reported that skin immunisation elicits a long-lived pool of memory CD8+ T cells in the FRT detected for up to 365 days post immunisation (Zaric M et al, JCI, 2017). Moreover, we have shown that antigen-specific CD8+ T cells recruited to the FRT express cell surface

markers (CD103, CD49a, CD69 and CXCR3) consistent with a tissue resident memory phenotype (Zaric M et al JCR 2017).

3. Given that the ability of the Ad5 virus to seed the FRT when administered intravaginally was dose dependent and that 3.5×10^2 vp could not elicit these CD8 T cells, whereas 1×10^9 vp did generate CD8 in FRT, this suggests that there could be a dose dependence in the skin as well? Do the authors have evidence for such a dose response?

To examine whether there is an Ad5 virus vector dose dependence in the skin that is necessary to prime and recruit CD8⁺ T cells to the FRT, groups of mice were skin immunised with Ad CN54 gag at a dose of either 1×10^7 vp or 1×10^9 vp. At day 14 post immunisation, Db/CN54 gag tetramer⁺ CD8⁺ T cells were measured in the spleen and FRT (new Supplementary Figure 2). We observe a significant decrease in frequency of Db/CN54 gag tetramer + CD8⁺ T cells (both in the FRT and spleen) after immunisation with the lower virus vector dose. This indicates a dose dependency in generating systemically primed CD8⁺ T cells after skin immunisation and in their recruitment to the FRT. We have revised the manuscript to include this point.

4. The authors showed the connection of induction of CXCL9 producing inflammatory Ly6C monocytes to aid in the recruitment of CXCR3⁺ CD8 T cells, but they did not address the contribution of the dendritic cells which can also express increased CXCL9. Do the DC themselves have a role in the recruitment?

This is an interesting question that forms the basis of on-going work to gain a deeper mechanistic insight into the pathways aiding recruitment of CXCR3⁺ CD8 T cells to the FRT. Our data suggests there is a complex loop of DC, ILC1 and Ly6C monocyte interactions that play a role in CXCL9 guided recruitment of CXCR3⁺ CD8⁺ T-cells. DCs and Ly6C monocytes likely mediate complementary functions in CXCL9 production that drive CXCR3⁺ CD8 T cell migration.

We observe that ablation of ILC1 cells by treatment with an anti-NK1.1 depleting antibody is associated with a significant decrease in recruitment of tetramer⁺ CD8⁺ T cells to the FRT (Figure 8c). As ILC1 cells are necessary for CXCR3⁺ CD8 T cell recruitment, we have now analysed the effect of ILC1 ablation on DCs that produce CXCL9. We find a significant decrease in frequency of DCs that express CXCL9 in the FRT of skin-immunised anti-NK1.1 Ab treated mice when compared with control treated mice ($p < 0.01$) (Figure R2 a and b). Additionally, the frequency of CXCL9⁺ DCs is significantly decreased in immunocompetent B6 and RAG-1- deficient mice treated with anti-IFN γ (Figure R3 a and b) consistent with ILC1 cell intrinsic IFN γ promoting CXCL9 production by DC that contributes to the CXCL9 chemokine microenvironment in the FRT necessary to recruit CXCR3⁺ CD8⁺ T cells.

[Redacted]

5- this data seems to particularly be important in protection by CD8 T cells, as the HPV prophylactic vaccines appear to protect from infection at the mucosa with high efficacy and these are applied by the IM route. These vaccines are however associated with Ab based protection. Perhaps the authors should consider mentioning this difference in the discussion section.

Thank you for the suggestion. We have added the following text into the discussion: “Our findings bare direct relevance to optimising delivery of vaccines where local immune surveillance by recruited CD8⁺ T cells plays a critical role in protection at the epithelial barrier (Hansen SG et al Nature, 2011; Hansen SG et al. Nature Medicine 2018). In contrast, existing vaccines that provide antibody-based protection at the cervico-vaginal epithelia, exemplified by the IM administered prophylactic HPV sub-unit vaccine, have been proposed to work by a different mechanism, via transudation or exudation of serum IgG as opposed to sub unit vaccination imprinting recruitment of antibody secreting cells to the epithelia (Day PM et al 2010)”.

There a few grammatical errors that should be addressed in the intro section and in the discussion.

We thank the reviewer for bringing this to our attention to improve the quality of our manuscript for the readership of the journal. Highlighted text in the revised manuscript indicates where grammatical changes have been made.

References

Barber DL, Wherry EJ, Ahmed R. Cutting edge: rapid in vivo killing by memory CD8 T cells. *J Immunol.* 171 (1): 27-31 (2003)

Nakanishi N et al. CD8(+) T lymphocyte mobilization to virus-infected tissue requires CD4(+) T-cell help. *Nature*, 462: 510-513 (2009)

Wang Y et al. Vaginal type-II mucosa is an inductive site for primary CD8⁺ T-cell mucosal immunity. *Nat. Commun.* 6:6100. doi: 10.1038/ncomms7100. (2015)

Cuburu N et al, Adenovirus vector-based prime-boost vaccination via heterologous routes induces cervicovaginal CD8⁺ T cell responses against HPV16 oncoproteins. *Int. J Cancer* 142(7): 1467-1479 (2018).

Kurachi M et al Chemokine receptor CXCR3 facilitates CD8(+) T cell differentiation into short-lived effector cells leading to memory degeneration. *J Exp Med.* 208 (8): 1605-1620 (2011).

Shah S et al. An extrafollicular pathway for the generation of effector CD8(+) T cells driven by the proinflammatory cytokine, IL-12. *eLife.* doi: 10.7554/eLife.09017 (2015).

Hu JK et al. Expression of chemokine receptor CXCR3 on T cells affects the balance between effector and memory CD8 T-cell generation. *PNAS.* 08(21):E118-27. doi: 10.1073 (2011).

Provine NM et al. Immediate Dysfunction of Vaccine-Elicited CD8+ T Cells Primed in the Absence of CD4+ T Cells. *J. Immunol.* 197(5):1809-22 (2016).

Chowell D et al. TCR contact residue hydrophobicity is a hallmark of immunogenic CD8+ T cell epitopes. *112(14):E1754-62 (2015).*

Zaric M et al. Long-lived tissue resident HIV-1 specific memory CD8+ T cells are generated by skin immunization with live virus vectored microneedle arrays. *JCR.* 268:166-175 (2017).

Hansen SG et al. Profound early control of highly pathogenic SIV by an effector memory T-cell vaccine. *Nature.* 473(7348):523-7 (2011).

Hansen SG et al. Prevention of tuberculosis in rhesus macaques by a cytomegalovirus-based vaccine. *Nature Med.* 24(2):130-143 (2018).

Day PM et al. In vivo mechanisms of vaccine-induced protection against HPV infection. *Cell Host Microbe,* 16;8(3):260-70 (2010).

REVIEWERS' COMMENTS:

Reviewer #1 (Remarks to the Author):

The authors have adequately addressed the points I raised. Some of the comments I raised were due to misunderstandings that have been clarified. The manuscript is substantially improved in its revised version.

Salvatore Valitutti

Reviewer #2 (Remarks to the Author):

Thank you for allowing me to review the revised manuscript by Prof Klavinskis and co-workers entitled "Skin immunisation activates an innate lymphoid cell-monocyte axis regulating CD8+ effector recruitment to mucosal tissues."

After reading over the comments and the response, I am satisfied that the authors have largely addressed my prior concerns as well as the critiques of reviewers overall.

Specifically Looking at the responses to Reviewer 1 . For comment 1, The authors showed that their study design rules out transferred CD45.1 cells that would recognize other antigens this is important.

Regarding Comment 2, The authors were transparent with the limitations of the target killing assay in that they did not isolate enough of these CD8 T cells from the FRT to analyze the killing with pure CD8 T cell at cognate target cell ratios.

Their specification that the gag peptide used to pulse the target cells and the Tetramer peptide are both to the immunodominant Db restricted gag epitopes alleviates some of the reviewer's concern that the target cells are not being killed by the Tet+ CD8s

For comment 3, the authors suggestion that having the anti-CXCR3 blocking antibody in the recipient mouse is a relevant and clearer answer as it allows them to circumvent a number of issues that blocking CXCR3 would have on the development of important CD8 T cells in the first place. The explanation of CXCR3 expression in Tet negative CD8s in naive mice is reasonable

Overall the referenced supplementary figure addresses the issue of CXCL9 expression from the Ly6C monocyte population at day 14 in the FRT.

Additional comments were addressed for the other reviewer. I'm satisfied that they looked at IM delivery for induction of the CXCR3+ CD8 T cell population.

It's interesting that they do see these cells with IM. This suggests that perhaps Ad vector is influencing this outcome.

for the second comment the prior study cited was important.

The response to comment 3, as was likely shows a dose dependent immune response, which has importance for future intradermal vaccine platform that may rely on this CXCR3 CXCL9 IFN γ axis as one mode of induced protection.

Comment 4 ,the data included that CXCL9+ DCs are decreased when the ICL1 cells are blocked lends additional credence this model as it supports the importance of ICL1 derived IFN γ suggesting that these two cell populations likely work in concert to recruit CXCR3+ T cells.

It appears based on their response that they intend to further flush out this pathway, which is important for future studies.

The authors should strengthen that this axis for AD was examined in most detail here, additional studies of additional forms of immunogens is also important.

REVIEWERS COMMENTS

Reviewer # 1 (Remarks to author)

The authors have adequately addressed the points I raised. Some of the comments I raised were due to misunderstandings that have been clarified. The manuscript is substantially improved in its revised version.

Salvatore Valitutti

Author response

We are pleased that we have conclusively clarified all the points raised and thank the reviewer for their time and kind attention to our manuscript.

Reviewer # 2 (Remarks to author)

Thank you for allowing me to review the revised manuscript by Prof Klavinskis and co-workers entitled "Skin immunisation activates an innate lymphoid cell-monocyte axis regulating CD8+ effector recruitment to mucosal tissues."

After reading over the comments and the response, I am satisfied that the authors have largely addressed my prior concerns as well as the critiques of reviewers overall.

Author response

We thank the reviewer for their time, attention and supportive comments to our revised manuscript.

Specifically looking at the responses to Reviewer 1.

For comment 1, The authors showed that their study design rules out transferred CD45.1 cells that would recognize other antigens this is important.

We agree.

Regarding Comment 2, The authors were transparent with the limitations of the target killing assay in that they did not isolate enough of these CD8 T cells from the FRT to analyze the killing with pure CD8 T cell at cognate target cell ratios.

Their specification that the gag peptide used to pulse the target cells and the Tetramer peptide are both to the immunodominant Db restricted gag epitopes alleviates some of the reviewer's concern that the target cells are not being killed by the Tet+ CD8s

We appreciate the reviewer's confirmation of the technical limitations working with cells isolated from the murine FRT and that our experimental design provides a definitive answer.

For comment 3, the authors suggestion that having the anti-CXCR3 blocking antibody in the recipient mouse is a relevant and clearer answer as it allows them to

circumvent a number of issues that blocking CXCR3 would have on the development of important CD8 T cells in the first place. The explanation of CXCR3 expression in Tet negative CD8s in naive mice is reasonable

We agree.

Overall the referenced supplementary figure addresses the issue of CXCL9 expression from the Ly6C monocyte population at day 14 in the FRT.

Thank you for confirming.

Additional comments were addressed for the other reviewer.

I'm satisfied that they looked at IM delivery for induction of the CXCR3+ CD8 T cell population. It's interesting that they do see these cells with IM. This suggests that perhaps Ad vector is influencing this outcome.

This is an avenue for future studies to drill down.

For the second comment the prior study cited was important.

Thank you for confirming.

The response to comment 3, as was likely shows a dose dependent immune response, which has importance for future intradermal vaccine platform that may rely on this CXCR3 CXCL9 IFN γ axis as one mode of induced protection.

We agree.

Comment 4, the data included that CXCL9+ DCs are decreased when the ICL1 cells are blocked lends additional credence this model as it supports the importance of ICL1 derived IFN γ suggesting that these two cell populations likely work in concert to recruit CXCR3+ T cells.

We agree.

It appears based on their response that they intend to further flush out this pathway, which is important for future studies.

That is correct, further studies are addressing this.

The authors should strengthen that this axis for Ad was examined in most detail here, additional studies of additional forms of immunogens is also important.

Thank you for the suggestion. We have added the following text into the original discussion:

“Although our study with Ad5 vectored vaccines was studied in most detail and emphasised local activation of innate NK 1.1⁺ cells and their role in orchestrating recruitment of CD8⁺ T effector and CD8⁺ T_{RM} cells to the FRT after skin immunisation, our findings do not preclude if this response state of NK1.1⁺ cells might also be dictated in part by cytokines acting systemically from the site of immunisation. Additional studies with other vaccine vectors and forms of immunogens will be important.